# Adversarial Locomotion and Motion Imitation for Humanoid Policy Learning

**Jiyuan Shi** [1*]   **Xinzhe Liu** [1,2*]   **Dewei Wang** [1,3*]   **Ouyang Lu** [1,4]   **Sören Schwertfeger** [2]
**Chi Zhang** [1]   **Fuchun Sun** [5]   **Chenjia Bai**[1†]   **Xuelong Li** [1†]

[1]Institute of Artificial Intelligence (TeleAI), China Telecom
[2]ShanghaiTech University     [3]University of Science and Technology of China
[4]Northwestern Polytechnical University     [5]Tsinghua University

## Abstract

Humans exhibit diverse and expressive whole-body movements. However, attaining human-like whole-body coordination in humanoid robots remains challenging, as conventional approaches that mimic whole-body motions often neglect the distinct roles of upper and lower body. This oversight leads to computationally intensive policy learning and frequently causes robot instability and falls during real-world execution. To address these issues, we propose Adversarial Locomotion and Motion Imitation (ALMI), a novel framework that enables adversarial policy learning between upper and lower body. Specifically, the lower body aims to provide robust locomotion capabilities to follow velocity commands while the upper body tracks various motions. Conversely, the upper-body policy ensures effective motion tracking when the robot executes velocity-based movements. Through iterative updates, these policies achieve coordinated whole-body control, which can be extended to loco-manipulation tasks with teleoperation systems. Extensive experiments demonstrate that our method achieves robust locomotion and precise motion tracking in both simulation and on the full-size Unitree H1-2 robot. Additionally, we release a large-scale whole-body motion control dataset featuring high-quality episodic trajectories from MuJoCo simulations. The project page is `https://almi-humanoid.github.io`.

## 1   Introduction

Humans exhibit diverse and expressive whole-body movements in various activities [1, 2]. For example, in dancing, humans depend on the lower body for stable movements and gait control, and the upper body executes precise actions to accomplish specific movements. This coordination allows for expressive and purposeful motions that are highly adaptable to different situations. However, achieving human-like whole-body coordination remains a highly challenging task for humanoid robots. Current methods employ motion re-targeting and Reinforcement Learning (RL) to learn a whole-body policy that can track human motions by taking the tracking errors as rewards and optimizing a whole-body policy to maximize such rewards [3, 4].

However, this kind of approach has significant limitations. First, due to the high number of Degrees of Freedom (DoF) in the humanoid robot, directly learning a whole-body control policy demands a complex reward structure and makes the training process highly expensive. Second, the differences between various motions, along with some human movements beyond a robot's physical capabilities, make it difficult for the RL policy to converge. In practice, since such whole-body learning approaches

---

*Equal Contribution    †Correspondence to: Chenjia Bai (baicj@chinatelecom.cn)

39th Conference on Neural Information Processing Systems (NeurIPS 2025).

prioritize precise motion tracking over balance maintenance, the policy often neglects the fundamental need for robot stability. Meanwhile, the poor stability of the lower body in turn affects the execution of upper-body motion and reveals challenges in real-world deployment, such as frequent robot falls. We identify that the main reason is that above methods do not separately consider the unique roles of the upper and lower bodies in motion learning, specifically, the lower body's role in robust locomotion and the upper body's task of precise motion imitation.

In this paper, we propose a novel framework, named *Adversarial Locomotion and Motion Imitation* (**ALMI**), that separately learns robust locomotion and motion imitation policies for the upper and lower bodies, respectively. Specifically, the lower body learns to follow different velocity commands while withstanding adversarial disturbances from the upper body, which leads to robust locomotion even when the upper-body movements significantly disrupt the stability. Conversely, the upper body learns to track reference motions accurately despite adversarial disturbances caused by lower-body instability, which leads to expressive motion imitation even when the lower body is commanded to move rapidly over uneven terrain. Through iterative updates of the upper and lower policies, these policies achieve coordinated whole-body control, which can be extended to loco-manipulation tasks with open-loop upper-body control via teleoperation systems. Unlike recent works [5–7] that adopt separate control for the upper and lower body, ALMI involves an adversarial training process to obtain coordinated behavior, which ensures the upper and lower policies converge to an equilibrium under mild conditions. Extensive experiments demonstrate that ALMI achieves robust locomotion and precise motion tracking in both simulation and on a full-size Unitree H1-2 humanoid robot.

Further, we construct a large-scale whole-body control dataset, named **ALMI-X**, featuring episodic trajectories for the Unitree H1-2 robot in MuJoCo simulations. Each episode is generated by the ALMI-acquired policy, where the lower body is controlled by the velocity command, and the upper body is controlled by joystick commands (i.e., desired joints) of the reference motion from the AMASS dataset [8]. We annotate language descriptions for each episode data according to both commands of the lower and upper bodies, e.g. *moving backward slowly and wave the left hand*. The ALMI-X dataset contains more than 80K trajectories with text commands and corresponding trajectories. We also give preliminary attempts to train a foundation model from the collected ALMI-X data, which serves an end-to-end whole-body control policy by leveraging a Transformer-based architecture.

Our main contributions are summarized as follows: (i) We propose a novel adversarial training framework (ALMI) for robust locomotion and precise motion imitation for humanoid robots, addressing the distinct roles of upper and lower body through separate policy learning. (ii) We create the first large-scale whole-body control dataset (ALMI-X) that integrates language descriptions with robot trajectories, facilitating the training of foundation models for humanoid whole-body control. (iii) We conduct extensive experiments to verify the effectiveness of the ALMI policy and the preliminary study for the humanoid foundation control model in both the simulation and the real world.

## 2 Preliminaries

In our work, we adopt an adversarial training framework for the policy learning of both the upper and lower bodies. Specifically, we consider an MDP $\mathcal{M} = (\mathcal{S}, \mathcal{A}^l, \mathcal{A}^u, T, \gamma, r^l, r^u, P)$, where the lower and upper bodies share the same state space $\mathcal{S}$ while having different action space, that is, $\mathcal{A}^l$ and $\mathcal{A}^u$, respectively. $r^l$ is the command-following reward for the locomotion policy (i.e., the lower body), and $r^u$ is the tracking reward for motion-imitation policy (i.e., the upper body). $P(s'|s, a^l, a^u)$ is the transition probability function that denotes the probability of transitioning to the next state $s'$ given the current state and the actions of both players. We aim to learn two policies, i.e., $\pi^l$ and $\pi^u$, that control different actions for the lower and upper bodies, respectively.

We adopted the full-sized Unitree H1-2 robot in our work, which has 27 DoFs in total, and the policy controls 21 DoFs, excluding the 3 DoFs in each wrist of the hands. The state space is defined as $s = (s_t^{\text{prop}}, c^l, \phi_t, g^u)$, where $s_t^{\text{prop}} = [q_t, \dot{q}_t, \omega_t, gv_t, a_{t-1}^l]$ is the proprioception of the robot, $c^l = [\hat{v}_{x,t}, \hat{v}_{y,t}, \hat{\omega}_{\text{yaw},t}]$ is the velocity command for the lower-body, and $\phi_t \in \mathbb{R}^2$ is the phase parameter at each time step, and $g^u \in \mathbb{R}^9$ is the reference joint position for the upper body. In proprioception, the notation $q_t \in \mathbb{R}^{21}, \dot{q}_t \in \mathbb{R}^{21}, \omega_t \in \mathbb{R}^3, gv_t \in \mathbb{R}^3, a_{t-1}^l \in \mathbb{R}^{12}$ is the joint position, the joint velocity, the angular velocity of the base, the projected gravity of the base, and the last action of the lower body, respectively.

For the lower body, the policy $\pi^l$ gives an action $\boldsymbol{a}^l \in \mathbb{R}^{12}$ representing target joint positions of the lower body according to the proprioception, commands, and phase variables at each step, which are fed into the PD controller to calculate the joint torques. For the upper body, policy $\pi^u$ takes proprioception and reference joint position as input, and the action $\boldsymbol{a}^u \in \mathbb{R}^9$ includes target positions of the 3 shoulder joints and 1 elbow joint per arm, along with the waist yaw joint. As a result, the two policies ($\pi^l$ and $\pi^u$) control distinct action spaces of the robot, and share the same state space by masking the irrelevant commands for the different policy.

## 3 Methods

In this section, we first present the theoretical foundation for the adversarial learning framework, and then give practical algorithms for implementing such a framework for humanoid robots.

### 3.1 The Adversarial Learning Framework

**Problem Formulation for learning $\pi^l$.** We consider a two-player zero-sum Markov game [9, 10] to learn a robust lower-body policy $\pi^l$. At each time step, the two players (i.e., $\pi^l$ and $\pi^u$) choose different actions ($a^l$ and $a^u$), and the humanoid robot executes both actions to obtain the reward and the next state. In learning $\pi^l$, we consider the lower body as *agent*, and the upper body is *adversary* that causes adversarial disturbances to the locomotion policy. Thus, the lower-body policy receives the command-following reward as $r^l(s, a^l, a^u)$, and the upper-body policy obtains a negative reward $-r^l(s, a^l, a^u)$. Formally, the value function $V^l(\pi^l, \pi^u)$ is defined as

$$V^l(s, \pi^l, \pi^u) := \mathbb{E}_{\pi^l, \pi^u} \left[ \sum_{t=0}^{T} r^l(s_t, a^l, a^u) \big| s_0 = s \right], \tag{1}$$

where $\mathbb{E}_{\pi^l, \pi^u}$ is the under the trajectory distribution induced by $\pi^l$ and $\pi^u$. Then we have the value function as $V_\rho^l(\pi^l, \pi^u) = \mathbb{E}_{s \sim \rho}[V^l(s, \pi^l, \pi^u)]$, which is defined by the expectation of accumulated locomotion reward $r^l$. Then the locomotion policy $\pi^l$ is learned to *maximize* the value function to better follow commands in locomotion, while the upper body policy tries to *minimize* the value function, aiming to provide an effective disturbance to help learn a robust locomotion policy. Formally, the two players form a Markov game and there exists a Nash equilibrium such have

$$V_\rho^l(\pi^{l*}, \pi^{u*}) = \max_{\pi^l} \min_{\pi^u} V_\rho^l(\pi^l, \pi^u). \tag{2}$$

To solve this max-min problem, we adopt an independent RL optimization process for both players. Specifically, the *agent* obtains $[(s_0, a_0^l, r_0^l), \ldots, (s_T, a_T^l, r_T^l)]$, and *adversarial* obtains $[(s_0, a_0^u, r_0^u), \ldots, (s_T, a_T^u, r_T^u)]$ by executing each policy in the game to sample a trajectory, where each player is oblivious to the actions of the other player. The two players optimize their policies independently with policy gradients using their own experiences. Then the following theorem holds.

**Theorem 3.1.** *Given $\epsilon > 0$, suppose each policy has $\varepsilon$-greedy exploration scheme with factors of $\varepsilon_x \asymp \epsilon$ and $\varepsilon_x \asymp \epsilon^2$, under a specific two-timescale rule of the two players' learning-rate following the independent policy gradient, we have*

$$\max_{\pi^l} \min_{\pi^u} V_\rho(\pi^l, \pi^u) - \mathbb{E} \left[ \frac{1}{N} \sum_{i=1}^{N} \min_{\pi^u} V_\rho(\pi^u, \pi^{l(i)}) \right] \leq \epsilon \tag{3}$$

*after $N$ episodes, which results in a $\epsilon$-approximate Nash equilibrium.*

Intuitively, the max-min game for the lower and upper bodies leads to a robust locomotion policy (by maximizing $V_\rho^l$ in the outer loop) even when the upper-body movements significantly disrupt the whole-body balance (by minimizing $V_\rho^l$ in the inner loop). This adversarial learning process is guaranteed to converge to a $\epsilon$-approximate Nash equilibrium according to Theorem 3.1.

**Problem Formulation for learning $\pi^u$.** Learning a precise motion imitation policy $\pi^u$ follows a similar process by considering $\pi^u$ as the *agent* and $\pi^l$ as the *adversary*. In the Markov game, the upper body receives a motion tracking reward $r^u(s, a^l, a^u)$ and the lower body receives $-r^u(s, a^l, a^u)$, which aims to give adversarial disturbances for motion tracking. Then we define $V^u(s, \pi^l, \pi^u) := \mathbb{E}_{\pi^l, \pi^u} \left[ \sum_{t=0}^{T} r^u(s_t, a^l, a^u) \big| s_0 = s \right]$ and $V_\rho^u(\pi^l, \pi^u) = \mathbb{E}_{s \sim \rho}[V^u(s, \pi^l, \pi^u)]$. Then the Markov game is formulated as

$$V_\rho^u(\pi^{l*}, \pi^{u*}) = \max_{\pi^u} \min_{\pi^l} V_\rho^u(\pi^l, \pi^u). \tag{4}$$

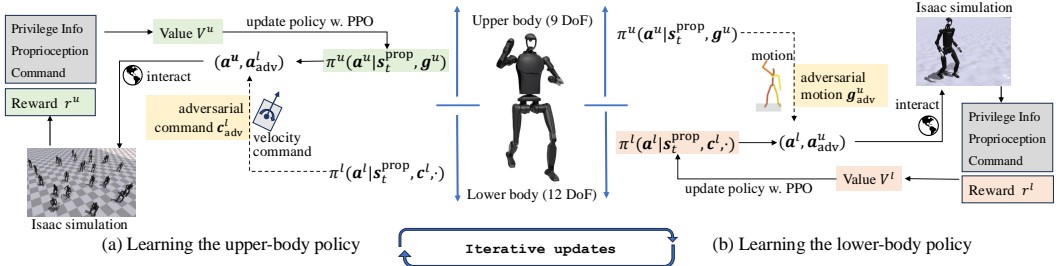

Figure 1: The overview of ALMI. (a) In updating the upper-body policy $\pi^u$, we sample adversarial velocity command $\boldsymbol{c}_{\text{adv}}^l$ and obtains $\boldsymbol{a}_{\text{adv}}^l$. Then we use $(\boldsymbol{a}^u, \boldsymbol{a}_{\text{adv}}^l)$ to interact with the environment to collect experiences, which are used to update $\pi^u$ via PPO algorithm [11]. (b) Similarly, in updating the lower-body policy, we sample adversarial motion $\boldsymbol{g}_{\text{adv}}^u$ and obtains $\boldsymbol{a}_{\text{adv}}^u$. Then we use $(\boldsymbol{a}^l, \boldsymbol{a}_{\text{adv}}^u)$ to interact and update $\pi^l$. The two policies $(\pi^l, \pi^u)$ finally converge via multiple mutual iterations.

By performing independent policy gradient optimization for the two players, we can prove that such a process also results in a $\epsilon$-approximate Nash equilibrium.

**Simplified Formulation.** According to Eq. (2) and (4), we require two pairs of $(\pi^l, \pi^u)$ to optimize $V^l$ and $V^u$ since they are defined by the opposite max-min problem, which could be computationally expensive. To address this, we propose a simplified framework by learning a single paired policy $(\pi^l, \pi^u)$. Specifically, (i) in learning the locomotion policy $\pi^l$, we optimize the parameters of $\pi^l$ while keeping the upper-body policy $\pi^u$ fixed. However, we sample an *adversarial motions* with a designed curriculum from the motion dataset, and set the corresponding reference joint position $\boldsymbol{g}_{\text{adv}}^u$ as the condition of $\pi^u(\cdot|\boldsymbol{s}_t^{\text{prop}}, \boldsymbol{g}_{\text{adv}}^u)$ to generate adversarial actions $\boldsymbol{a}_{\text{adv}}^u$. Similarly, (ii) in learning the motion imitation policy $\pi^u$, we only update $\pi^u$ and keep the lower-body policy $\pi^l$ fixed. Then, we sample an *adversarial command* $\boldsymbol{c}^l$ with the curriculum for the locomotion policy $\pi^l(\cdot|\boldsymbol{s}_t^{\text{prop}}, \boldsymbol{c}^l, \cdot)$ to generate $\boldsymbol{a}_{\text{adv}}^l$. To conclude, we change the inner-loop optimization of the max-min problem (in Eq. (2) and (4)) from the parameter space to the command space (i.e., by sampling $\boldsymbol{g}_{\text{adv}}^u$ and $\boldsymbol{c}_{\text{adv}}^l$), which leads to a practical efficient algorithm. Fig. 1 gives an overview of our method.

### 3.2 Robust Locomotion for the Lower Body

Achieving robust locomotion in humanoid robots remains a significant challenge, primarily due to the dynamic nature of environments and the inherent complexity of multi-joint floating base systems. This instability is particularly pronounced in full-sized humanoid robots, which often require increased load capacity to execute whole-body control or loco-manipulation tasks. Such requirements typically lead to higher arm mass and inertia, exacerbating system instability. Consequently, during coordinated upper and lower body movements or mobile manipulation, disturbances generated by the upper body exert significant impacts on the lower body, necessitating robust policies to mitigate these effects.

Table 1: Reward terms and weights for lower policy.

| Term | Expression | Weight |
|------|------------|--------|
| | Penalty | |
| DoF position limits | $\mathbb{I}(\boldsymbol{q}^l \notin [\boldsymbol{q}_{\min}^l, \boldsymbol{q}_{\max}^l])$ | -5.0 |
| Alive | $\mathbb{I}(\text{robot stays alive})$ | 0.15 |
| | Regularization | |
| Linear velocity of Z axis | $\|\boldsymbol{v}_z\|_2^2$ | -2 |
| Angular velocity of X&Y axis | $\|\boldsymbol{w}_{\text{xy}}\|_2^2$ | -0.5 |
| Orientation | $\|\boldsymbol{g}_{\text{xy}}^{\text{root}}\|_2^2$ | -1 |
| Torque | $\|\boldsymbol{\tau}^l\|_2^2$ | -1e-5 |
| Base Height | $\|h - h^{\text{target}}\|_2^2$ | -10.0 |
| DoF acceleration | $\|\ddot{\boldsymbol{q}}^l\|_2^2$ | -2.5e-7 |
| DoF velocity | $\|\dot{\boldsymbol{q}}^l\|_2^2$ | -1e-3 |
| Lower body action rate | $\|\boldsymbol{a}_t^l - \boldsymbol{a}_{t-1}^l\|_2^2$ | -0.01 |
| Hip DoF position | $\|\boldsymbol{q}^{\text{hip roll\&yaw}}\|_2^2$ | -1 |
| Slippage | $\|\boldsymbol{v}_{\text{xy}}^{\text{feet}}\|_2^2 \times \mathbb{I}(\|F^{\text{feet}}\|_2 \geq 1)$ | -0.2 |
| Feet swing height | $\|\boldsymbol{q}_z^{\text{feet}} - 0.08\|_2^2 \times \mathbb{I}(\|F^{\text{feet}}\|_2 < 1)$ | -20 |
| Feet Contact | $\sum_{i=1}^{N_{\text{feet}}} \neg(\mathbb{I}(\|\boldsymbol{F}_{z,i}^{\text{feet}}\|_2 > 1) \oplus \mathbb{I}(\phi_i < 0.55))$ | 0.18 |
| Feet distance | $\exp(-100 \times d_{\text{feet}}^{\text{out of range}})$ | 0.5 |
| Knee distance | $\exp(-100 \times d_{\text{knee}}^{\text{out of range}})$ | 0.4 |
| Stand still | $\|\boldsymbol{q}_t^l - \boldsymbol{q}_{t-1}^l\|_2^2 \times \mathbb{I}(\|\boldsymbol{c}^l\|_2 < 0.1)$ | -2 |
| Ankle torque | $\|\boldsymbol{\tau}^{\text{ankle}}\|_2^2$ | -5e-5 |
| Ankle action rate | $\|\boldsymbol{a}_t^{\text{ankle}} - \boldsymbol{a}_{t-1}^{\text{ankle}}\|_2^2$ | -0.02 |
| Stance base velocity | $\|\boldsymbol{v}\|_2^2 \times \mathbb{I}(\|\boldsymbol{c}^l\|_2 < 0.1)$ | -1 |
| Feet contact forces | $\min(\|F^{\text{feet}} - 100\|_2^2, 0)$ | -0.01 |
| | Task | |
| Linear velocity | $\exp(-4\|\boldsymbol{c}_{\text{xy}} - \boldsymbol{v}_{\text{xy}}\|_2^2)$ | 2 |
| Angular velocity | $\exp(-4\|\boldsymbol{c}_{\text{yaw}} - \boldsymbol{v}_{\text{yaw}}\|_2^2)$ | 1 |

**Motivation of Curriculum.** In learning a robust locomotion policy, an adversarial upper body component is essential to generate perturbations. However, directly training such an adversary is challenging. Empirical results show that without constraints, the upper body can terminate episodes early or collide with the lower body to minimize its reward. Manually designing constraints is labor intensive, and the lower body can exploit flaws in the reward function, leading to poor real-world performance. Thus, as we discussed in §3.1, we simplify the original problem by sampling adversarial motions without updating the upper-body policy. Although aggressive sampling of highly dynamic motions can reduce the locomotion policy's value function (i.e. $V^l$) as in Eq. (1), this strategy cannot adequately teach the locomotion policy how to resist disturbances when the policy is relatively weak, resulting in slow convergence. To overcome this, we introduce a novel curriculum mechanism that starts with moderate disturbances and gradually increases their intensity as the lower body's robustness improves, ensuring efficient learning and robustness improvement throughout training.

Table 2: Reward terms and weights for upper policy.

| Term | Expression | Weight |
|---|---|---|
| | Penalty | |
| DoF position limits | $\mathbb{I}(\boldsymbol{q}^{\mathrm{u}} \notin [\boldsymbol{q}^{\mathrm{u}}_{\min}, \boldsymbol{q}^{\mathrm{u}}_{\max}])$ | -5.0 |
| Alive | $\mathbb{I}(\text{robot stays alive})$ | 0.15 |
| | Regularization | |
| Orientation | $\|\boldsymbol{g}^{\mathrm{root}}_{\mathrm{xy}}\|^2_2$ | -1 |
| Torque | $\|\boldsymbol{\tau}^{\mathrm{u}}\|^2_2$ | -1e-5 |
| Upper DoF acceleration | $\|\ddot{\boldsymbol{q}}^{\mathrm{u}}\|^2_2$ | -2.5e-7 |
| Upper DoF velocity | $\|\dot{\boldsymbol{q}}^{\mathrm{u}}\|^2_2$ | -1e-3 |
| Upper body action rate | $\|\boldsymbol{a}^{\mathrm{u}}_t - \boldsymbol{a}^{\mathrm{u}}_{t-1}\|^2_2$ | -0.01 |
| | Task | |
| Upper DoF position | $\exp(-0.5\|\hat{\boldsymbol{q}}^{\mathrm{u}} - \boldsymbol{q}^{\mathrm{u}}\|^2_2)$ | 10 |

**Dual Curriculum Mechanism.** We propose a novel dual curriculum mechanism for adversarial motion sampling by scoring upper-body motions based on their impact on the lower-body's stability. Specifically, we initially train a basic locomotion policy $\pi^l_0$ without any upper body intervention. Then we use a PD controller (in the first round) or $\pi^u$ to control the upper-body to track motions from the re-targeted AMASS [8] dataset in simulation, while the lower body follows the basic velocity commands. The robot inevitably falls initially due to upper body perturbations, and we record *survival length* (denoted as $l^{\mathrm{sl}} \in [0, l^{\mathrm{sl}}_{\max}]$) as the primary metric for motion difficulty. Then, motions are sorted into a list $\mathbf{M} = [M_1, \ldots, M_{|\mathbf{M}|}]$ by increasing difficulty based on survival length. Further, we introduce a factor, *motion scale* $\alpha_s \in [0, 1]$, to scale the target joint positions of the upper body as

$$\boldsymbol{q}^i_{\mathrm{target}} = \boldsymbol{q}^i_0 + \alpha_s(\boldsymbol{q}^i_{\mathrm{ref}} - \boldsymbol{q}^i_0). \tag{5}$$

For each iteration, the robot is assigned a window of motions $[M_{\alpha_d}, M_{\alpha_d+w}]$ from $\mathbf{M}$, where $w$ is the window size and $\alpha_d = 1$ at the start. These motions, scaled by $\alpha_s$, are used to control the upper body to provide disturbances. The lower-body policy $\pi^l_0$ is updated to $\pi^l_1 \leftarrow \pi^l_0$ through an RL optimization process with locomotion-related rewards detailed in Table 1. To adaptively adjust the difficulty of sampled motions, we calculate *mean survival length* (denoted as $l^{\mathrm{msl}}$) of the sampled motions as a metric to update $\alpha_d$ and $\alpha_s$, which progressively increases motion difficulty based on the current policy's anti-disturbance capability. Formally,

$$\alpha_d \leftarrow \begin{cases} \min(\alpha_d + w, |\mathbf{M}| - w), & \text{if} \quad l^{\mathrm{msl}} \geq 0.8 * l^{\mathrm{sl}}_{\max} \\ \max(\alpha_d - 2w, 0), & \text{otherwise} \end{cases}, \tag{6}$$

which increases the difficulty if $\pi^l$ can effectively resist the interference caused by sampled motions, and decreases the difficulty otherwise. The update of motion scale follows a similar process as

$$\alpha_s \leftarrow \begin{cases} \min(\alpha_s + 0.05, 1), & \alpha_d = |\mathbf{M}| - w \\ \max(\alpha_s - 0.01, 0), & \alpha_d = 0 \end{cases}, \tag{7}$$

which signifies that the motion scale $\alpha_s$ increases after all motions at the current scale are successfully managed. However, increasing $\alpha_s$ typically decreases $l^{\mathrm{msl}}$, potentially causing $\alpha_d$ to decrease and resulting in motion sampling that reverts to previous entries in $\mathbf{M}$. This triggers a new cycle of policy updates with adjusted motions and scales. To maintain relevance, the motion list $\mathbf{M}$ is periodically re-sorted to ensure alignment with the evolving robustness of the locomotion policy. The Algorithm 1 in Appendix B.2 gives a training process for the locomotion policy.

## 3.3 Motion Tracking for the Upper Body

The upper-body policy $\pi^u$ aims to track various motions under disturbances from lower-body movements with adversarial commands. Before motion tracking, we follow the process proposed by

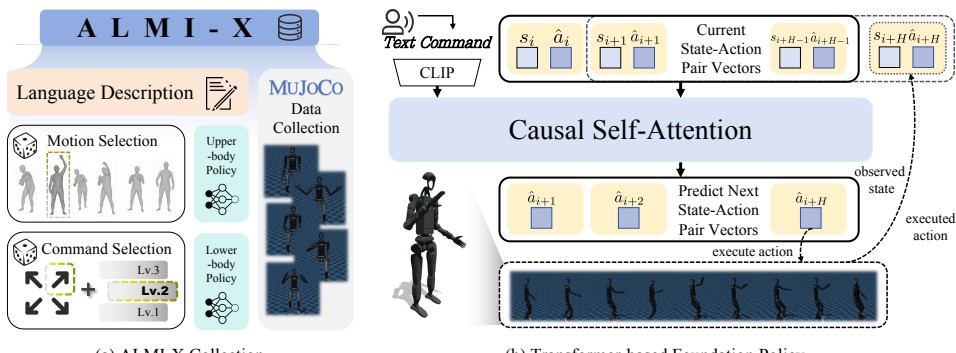

(a) ALMI-X Collection

(b) Transformer-based Foundation Policy

Figure 2: ALMI-X dataset and foundation model training. (a) ALMI-X features motion target and velocity command for the learned policies, combining with language description. (b) The foundation model learns $P(\hat{a}_{i+1}|s_{\leq i}, a_{\leq i}, \mathcal{T})$ from a segment of state-action pairs via causal attention. In inference, the last action is executed based on the history and obtains the true state for next steps.

PHC [12] to re-target the human motion from AMASS [8] dataset to humanoid robot. To enhance motion smoothness, we applied low-pass filtering to the re-targeted motion data.

In the adversarial training process, we use the locomotion policy $\pi^l$ in the current round to control the lower body, and the upper-body policy is learned by maximizing the motion tracking reward. Different from §3.2 that requires delicate curriculum mechanism, the curriculum for adversarial $\pi^l$ can be relatively simple since the command $\mathbf{c}^l$ only contains velocity commands. Specifically, $\mathbf{c}^l$ is sampled from a range of $[\mathbf{c}^l_{\min}, \mathbf{c}^l_{\max}]$, and we adjust the value of $\mathbf{c}^l_{\min}$ and $\mathbf{c}^l_{\max}$ according to the tracking error of motions in the upper body, which is affected by the movement of the lower body. In training $\pi^u$, the lower-body command gradually increases its difficulty based on the tracking error of the upper body. The reward design for the upper-body policy is described in Table 2, and the detailed curriculum mechanism is provided in Appendix B.3.

## 4   ALMI-X Datasets and Foundation Model

The adversarial training processes outlined in §3.2 and §3.3 involve an iterative procedure over multiple epochs to reach convergence. The final policies are used to gather a large-scale high-quality dataset, which is used to train a foundation model dedicated to end-to-end whole-body control.

**Dataset Construction.**   We collect ALMI-X dataset in MuJoCo simulations by running the trained ALMI policy, which consists of the final policy $\pi^l$ and $\pi^u$. In this simulation, we combine a diverse range of upper-body motions with omnidirectional lower-body commands, and employ a pre-defined paradigm to generate corresponding linguistic descriptions for each combination. (i) For the upper-body, we collect data using our upper-body policy to track various motions from a subset of the AMASS dataset [8], where we remove entries with indistinct movements or those that could not be matched with the lower-body commands, such as *push from behind*. (ii) For the lower-body, we first categorize command directions into several types according to different combination of linear and angular velocity command and define 3 difficulty levels for command magnitudes, then the lower-body command is set by combining direction types and difficulty levels. Overall, each upper-body motion from the AMASS subset is paired with a specific direction type and a difficulty level serving as the inputs of $\pi^u$ and $\pi^l$ to control the robot. In addition, trajectories in which the lower body *keep standing* while the upper body tracks motions are also incorporated into the dataset. Each language description $\mathcal{T}$ in ALMI-X is organized as *"${movement mode} ${direction} ${velocity level} and ${motion}"*, each of which corresponds to the data collected from a trajectory lasting on average 4 seconds with 200 steps. For each $\mathcal{T}$, we run two policies (i.e., $\pi^u$, $\pi^l$) based on the commands obtained from the aforementioned combinations to achieve humanoid whole-body control. We record the trajectory information such as the robot states $\tau_s = (s_0, s_1, ..., s_T)$ and actions $\tau_a = (a_0, a_1, ..., a_T)$. Our data exhibits high executability, and the details are discussed in Appendix C.

**Foundation Model for Humanoid Control.**   Using the ALMI-X dataset, we give preliminary attempts to train a whole-body foundation model with supervised learning that can execute various

motions in response to text commands. We adopt a Transformer decoder architecture that implements causal self-attention modules over a text command and historical state-action pairs. Unlike UH-1 [13] that predicts the entire action sequence based on the text command, our policy autoregressively models the next action by incorporating real-time interaction history between the robot and the environment. This architecture avoids executing the entire action sequence without considering the random noise and interference of the environment, instead adjusting actions based on intermediate states, ensuring robust and adaptive control. Moreover, compared to methods focused solely on walking [14], ALMI-X contains more diverse data from various motions and velocity commands, enabling the learning of the foundation model to achieve complex language-guided whole body control. The architecture is shown in Fig. 2 and the details are given in Appendix D.

## 5 Related Works

**Humanoid Locomotion.** Traditional humanoid locomotion primarily focuses on trajectory planning [15, 16] and Zero-Moment Point (ZMP)-based gait synthesis [17, 18]. Other methods like whole-body control [19, 20] and model-predictive control [21, 22] gain significant progress with dynamics modeling, while it can be difficult to model rich contact in complex terrains [23]. Recently, learning-based algorithms have shown promising results in humanoid locomotion tasks [24, 25]. RL has become a powerful tool for learning locomotion policies by enabling agents to interact with the environment through parallelized simulations [26]. Based on this, humanoids can acquire a wide range of skills, including walking on complex terrains [27], [28], gait control [29], standing up [30, 31], narrow-terrain navigation [32], jumping [33], and even parkour [34, 35]. However, most methods focus solely on controlling the lower-body joints, treating the upper body as a fixed load [36, 37]. Although some studies have explored whole-body control policies [32, 29], their primary objective is to learn an upper-body swinging policy to coordinate with the locomotion task. In contrast, our method enables the lower-body policy to resist disturbances caused by various upper-body motions.

**Humanoid Motion Imitation.** Recent research has proposed expressive motion imitation algorithms that learn human-like behaviors from motion capture datasets [38–40]. These methods employ motion re-targeting and RL-based optimization to achieve fine-grained motion imitation and postural stability [41–43]. However, balancing these two objectives can be challenging, as the execution of motions may disturb stability, while poor stability, in turn, affects the precision of motion execution. Several methods address this issue through careful reward design [44, 45], decomposed tracking strategies [6, 4, 46, 47], and residual learning [48]. Unlike these methods, our approach focuses solely on tracking human motions in the upper body. Although this may result in less expressiveness, it leads to robust locomotion in the lower body and precise motion tracking in the upper body, leading to an effective whole-body policy for real-world deployment. Other methods [49, 50, 5] also adopt a separate control paradigm for training the two parts with different rewards, while we employ an adversarial training process to promote coordinated behaviors between locomotion and motion imitation policies.

**Foundation Model for Robotics.** Our work is also related to the learning of foundation models for robotics. Previous works mainly focus on building vision-language-action models for end-to-end manipulation, such as MT-Diff [51], RT-1 [52], Octo [53], OpenVLA [54], RDT [55], $\pi_0$ [56], GO-1 [57], and BRS [58]. They rely on large-scale datasets collected by humans, such as RH20T [59], Bridge Data [60], RoboMIND [61], Open-X [62], AgiBot Data [57] and HGen-Bench[63]. However, these datasets focus mainly on arm manipulation with a fixed or wheeled platform. In contrast, our dataset (i.e., ALMI-X) targets whole-body control in humanoid robots, which involves controlling more DoFs than manipulation tasks. UH-1 [13] is closely related to ALMI, but learns a mapping from text descriptions to keypoints, and low-level actions are found to be unexecutable in real robots.

## 6 Experiments

In this section, we evaluate the performance of ALMI using the Unitree H1-2 robot in both simulated and real-world environments. Our experiments aim to address the following research questions: **Q1.** How does ALMI perform in terms of **tracking precision** for lower-body velocity commands and upper-body motions? **Q2.** How does ALMI perform regarding **stability** and **robustness** in complex scenarios? **Q3.** What benefits does **adversarial iterative training** and the **arm curriculum** mechanism provide for policy optimization? **Q4.** How does ALMI perform in the **real world**?

**Training Details.** Policy training is conducted within the Isaac Gym simulator utilizing 4,096 parallel environments. We perform three rounds of adversarial iterations and evaluate the policies of the final round. Notably, during the adversarial iterative training process, the initial lower-body policy converges after approximately $10^4$ steps. As iterations progress, the number of steps required for convergence decreases significantly. The total duration of the three training iterations is approximately 17 hours. For each baseline method, we use the checkpoint obtained after training convergence for testing. To enhance generalization, we implement domain randomization for sim-to-sim and sim-to-real transfers, with specific details provided in Appendix B.5.

**Evaluation Metrics.** For simulated evaluation, we adopt the high-quality CMU MoCap dataset [64] with 1122 motion clips (denoted as $\mathcal{D}_{\mathrm{cmu}}$) to evaluate different metrics in IsaacGym [26]. The metrics include (i) the *mean linear velocity error* $E_{\mathrm{vel}}(\mathrm{m/s})$ and the mean angular velocity error $E_{\mathrm{ang}}(\mathrm{rad/s})$ that evaluate the command tracking accuracy; (ii) the upper body *joints position error* $E_{\mathrm{jpe}}^{\mathrm{upper}}(\mathrm{m})$ and *key points error* $E_{\mathrm{kpe}}^{\mathrm{upper}}(\mathrm{m})$ that evaluates the motion tracking accuracy; (iii) the *joint difference* for both parts $E_{\mathrm{action}}^{\mathrm{upper}}(\mathrm{rad})$, $E_{\mathrm{action}}^{\mathrm{lower}}(\mathrm{rad})$; (iv) and the *projected gravity* $E_{\mathrm{g}}$ evaluate the stability of policy. (v) We also statistically analyze *survival rate* to assess the robustness of the policy.

**Baselines.** Our baselines are as follows: (i) **Exbody** [6], which employs a unified policy to control whole-body joints, tracking upper-body movements from motion data and lower-body root motion via command sampling. (ii) **ALMI (whole)**, which uses a single policy to control the upper and lower bodies, with identical reward functions for training each part as in ALMI. (iii) **ALMI (w/o curriculum)**. An ablation study that omits the arm curriculum, where the policy loads the motion randomly and the motion scale $\alpha_s$ is set to 1. (iv) **ALMI (w/o adv. learning)**. This study evaluates the impact of our iterative adversarial training by testing policies from the first and second rounds. We also compare our method with approaches that simultaneously imitate both upper and lower body movements. We consider two state-of-the-art methods: (v) **ExBody2** [4], which leverages a privileged teacher policy to distill precise mimicry skills into a student policy, enabling whole-body imitation, and (vi) **OmniH2O** [45], an imitation-based whole-body control method that supports real-time input from various input devices.

### 6.1 Main Result of ALMI

To evaluate locomotion and motion tracking capabilities, we calculate metrics in $\mathcal{D}_{\mathrm{cmu}}$ using commands in the lower body categorized into difficulty levels *easy*, *medium*, and *hard*, as described in Table 3. These levels encompass linear velocities in the $x$ and $y$ directions, angular velocity in the yaw direction, terrain level in the Isaac Gym and the presence of external forces. To control variables and facilitate analysis, we assess linear and angular velocities separately (setting the other to zero during testing) and report the average metrics.

To answer **Q1** and **Q2**, we give the comparative results in Table 4. (i) **Tracking accuracy.** ALMI demonstrates superior tracking precision across all difficulty levels for both

Table 3: Locomotion difficulty level setting.

| Level | $\hat{v}_{\mathrm{x,t}}$ | $\hat{v}_{\mathrm{y,t}}$ | $\hat{\omega}_{\mathrm{yaw,t}}$ | terrain level | push robot |
|---|---|---|---|---|---|
| | | | **Command & Environment** | | |
| easy | 0.7 | 0.0 | 0.2 | 0 | ✗ |
| medium | 1.0 | 0.3 | 0.4 | 3 | ✓ |
| hard | 1.3 | 0.6 | 0.6 | 6 | ✓ |

Table 4: Simulated evaluation of ALMI, ALMI (whole body) and Exbody on CMU dataset.

| Method | $E_{\mathrm{vel}}\downarrow$ | $E_{\mathrm{ang}}\downarrow$ | $E_{\mathrm{jpe}}^{\mathrm{upper}}\downarrow$ | $E_{\mathrm{kpe}}^{\mathrm{upper}}\downarrow$ | $E_{\mathrm{action}}^{\mathrm{upper}}\downarrow$ | $E_{\mathrm{action}}^{\mathrm{lower}}\downarrow$ | $E_{\mathrm{g}}\downarrow$ | Survival $\uparrow$ |
|---|---|---|---|---|---|---|---|---|
| | | | | **Metrics** | | | | |
| *Easy* | | | | | | | | |
| ALMI | **0.1135** | **0.2647** | **0.1931** | **0.0460** | **0.0462** | **0.0170** | **0.6919** | **1.0000** |
| ALMI(whole) | 0.1386 | 0.5433 | 0.5756 | 0.0704 | 0.0800 | 3.0356 | 0.9675 | 0.9991 |
| Exbody | 0.2383 | 0.4056 | 0.3559 | 0.0995 | 1.7813 | 1.8152 | 0.9693 | 0.8912 |
| *Medium* | | | | | | | | |
| ALMI | **0.2192** | **0.3520** | **0.2007** | **0.0450** | **0.0598** | **0.0172** | **0.7604** | **0.9852** |
| ALMI(whole) | 0.2380 | 0.5563 | 0.6734 | 0.0637 | 0.0409 | 2.9225 | 1.0750 | 0.9763 |
| Exbody | 0.3063 | 0.5087 | 0.3658 | 0.1233 | 1.7683 | 1.8019 | 1.0166 | 0.8845 |
| *Hard* | | | | | | | | |
| ALMI | **0.2202** | **0.4812** | **0.2116** | **0.0458** | **0.0600** | **0.0175** | **0.8551** | **0.9723** |
| ALMI(whole) | 0.3178 | 0.7224 | 0.7022 | 0.0635 | 0.0519 | 2.9317 | 1.1656 | 0.9491 |
| Exbody | 0.4838 | 0.5753 | 0.3758 | 0.1269 | 1.7352 | 1.7689 | 1.0243 | 0.8778 |

Table 5: Comparison with Exbody2 and OmniH2O on G1 in simulation.

| Method | $E_{\text{vel}} \downarrow$ | $E_{\text{ang}} \downarrow$ | $E_{\text{jpe}}^{\text{upper}} \downarrow$ | $E_{\text{kpe}}^{\text{upper}} \downarrow$ | $E_{\text{action}}^{\text{upper}} \downarrow$ | $E_{\text{action}}^{\text{lower}} \downarrow$ | $E_{\text{g}} \downarrow$ | Survival $\uparrow$ |
|--------|------|------|------|------|------|------|------|------|
| ALMI | **0.1396** | **0.2776** | **0.2367** | **0.0411** | **0.0198** | **0.7411** | **0.0977** | **0.9484** |
| OmniH2O | 0.1615 | 0.4166 | 1.0826 | 0.0598 | 1.2773 | 2.3219 | 0.1696 | 0.3882 |
| Exbody2 | 0.4015 | 0.6066 | 0.3821 | 0.0719 | 1.1797 | 1.3547 | 0.3367 | 0.8848 |

upper-body motions and lower-body velocity commands, consistently outperforming other baselines. The results highlight that a single policy struggles with upper-lower body coordination, limiting tracking accuracy. In contrast, our adversarial training framework simultaneously enhances the accuracy of both policies for their respective objectives. (ii) **Stability and robustness.** According to the survival rate, it can be found that with an improvement in difficulty levels, ALMI can track motions and velocity commands robustly and effectively prevent falls. This stability is attributed to the lower-body policy, which effectively learns to execute stable movements despite adversarial upper-body interference, underscoring the efficacy of adversarial training strategy.

As OmniH2O and Exbody2 are imitation-based whole-body control methods without velocity tracking, for ALMI, we use linear and angular velocity provided by the reference motion as velocity tracking command; for OmniH2O and Exbody2, they directly track the reference motions. This experiment is conducted without terrain or push. To indicate that our method can be extended to other robot platforms, we use Unitree G1 robot in this setting. The results are shown in 5. It can be observed that our method generalizes well across different robotic platforms and demonstrates superior robustness compared to imitation-based whole-body control methods.

To answer **Q3**, we evaluate the impact of our arm curriculum and adversarial learning techniques, with results presented in Table 6. We use the trained lower-body policies across three iterations, labeled *lower-1*, *lower-2*, and *lower-3*. Under easy velocity commands without environmental disturbances, performance differences among the iterations are minimal. However, as task complexity increased, *lower-3* demonstrated superior performance across all metrics compared to earlier iterations, while *lower-1* showed notably inferior results. During adversarial training, the lower and upper policies iteratively refined their motion and command tracking capabilities while generating progressively larger disturbances for each other, guided by the designed curriculum. For instance, to achieve high-velocity commands, the lower body might execute large-amplitude actions, inducing oscillations that disrupt upper-body movements, and vice versa. Through this iterative process, both policies gradually adapt to adversarial perturbations, eventually reaching a stable condition. In addition, Table 6 underscores the critical role of the arm curriculum. By systematically introducing upper-body motions from easy to hard in training, the lower-body policy effectively learned to handle increasingly difficult perturbations and motion scales.

Table 6: Ablation studies of adversarial training technique and arm curriculum in ALMI.

| Method | $E_{\text{vel}} \downarrow$ | $E_{\text{ang}} \downarrow$ | $E_{\text{jpe}}^{\text{upper}} \downarrow$ | $E_{\text{kpe}}^{upper} \downarrow$ | $E_{\text{action}}^{\text{upper}} \downarrow$ | $E_{\text{action}}^{\text{lower}} \downarrow$ | $E_{\text{g}} \downarrow$ | Survival $\uparrow$ |
|--------|------|------|------|------|------|------|------|------|
| Easy | | | | | | | | |
| lower-3 + upper-2 | **0.1135** | **0.2647** | 0.1931 | **0.0460** | **0.0462** | 0.0170 | 0.6919 | **1.0000** |
| lower-2 + upper-2 | 0.1164 | 0.2669 | 0.1955 | 0.0452 | 0.0475 | **0.0171** | 0.7121 | **1.0000** |
| lower-1 + upper-2 | 0.1271 | 0.2738 | 0.1928 | 0.0526 | 0.0642 | **0.0171** | 0.7052 | **1.0000** |
| w/o arm curriculum | 0.1411 | 0.2726 | **0.1924** | 0.0504 | 0.0618 | **0.0172** | 0.7472 | 0.9995 |
| Medium | | | | | | | | |
| lower-3 + upper-2 | **0.2192** | **0.3520** | **0.2007** | **0.0450** | **0.0598** | **0.0172** | **0.7604** | **0.9852** |
| lower-2 + upper-2 | 0.2213 | 0.3571 | 0.2032 | 0.0458 | 0.0607 | **0.0172** | 0.7748 | 0.9772 |
| lower-1 + upper-2 | 0.2262 | 0.3872 | 0.2173 | 0.0492 | 0.0604 | 0.0175 | 0.7730 | 0.9273 |
| w/o arm curriculum | 0.2571 | 0.4348 | 0.2068 | 0.0476 | 0.0601 | 0.0173 | 1.0587 | 0.9652 |
| Hard | | | | | | | | |
| lower-3 + upper-2 | **0.2202** | **0.4812** | **0.2116** | **0.0458** | **0.0600** | **0.0175** | **0.8551** | **0.9723** |
| lower-2 + upper-2 | 0.2892 | 0.5395 | 0.2231 | 0.0482 | 0.0645 | 0.0178 | 0.9479 | 0.9233 |
| lower-1 + upper-2 | 0.2566 | 0.5172 | 0.2451 | 0.0537 | 0.0777 | 0.0179 | 0.9462 | 0.8743 |
| w/o arm curriculum | 0.3658 | 0.6398 | 0.2394 | 0.0461 | 0.0726 | 0.0180 | 1.2042 | 0.8480 |

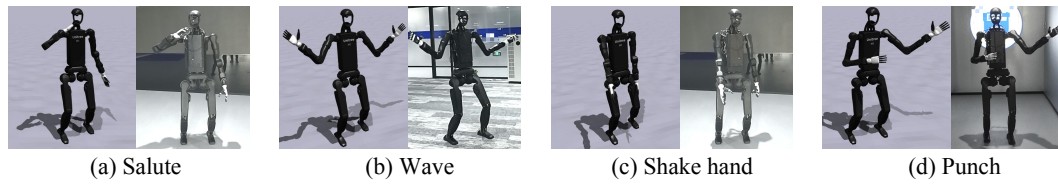

| (a) Salute | (b) Wave | (c) Shake hand | (d) Punch |

Figure 3: The sim-to-real comparison of humanoid robot in tracking various motions.

## 6.2 Result of Foundation Model

The Transformer-based foundation model leverages textual descriptions of the whole-body motion along with historical information to predict future action. In experiments, we find that when training on a subset of the ALMI-X dataset (e.g., waving-related data), the model converges to a robust policy, enabling execution of various waving and moving motions in the MuJoCo simulations. We highlight that this is particularly challenging, as no prior works show a purely offline policy can control full-sized humanoid robots to generate language-guided behaviors. However, when we extend the training data to the complete ALMI-X, the performance degraded, indicating limitations of the current model in handling highly diverse behaviors. ALMI-X dataset and our finding provide a promising starting point for future exploration, with detailed results provided in Appendix D.

## 6.3 Real-World Experiments

To answer **Q4**, we deploy ALMI on the Unitree H1-2 robot to evaluate real-world performance. For lower-body control, we use a joystick-mounted remote controller to send velocity commands to $\pi^l$, which controls the robot to achieve omnidirectional movement. For upper body control, we can adopt open-loop controller or ALMI policy to track various motions, as well as using VR devices to support dexterous control in loco-manipulation tasks. The details are given in Appendix F.

To evaluate the capabilities in maintaining stability in movements while tracking motions precisely, we control the robot to execute various upper body motions while standing or walking in omnidirection. Fig. 3 illustrates the motion tracking behavior, emphasizing the alignment between simulation and real-world performance. To further demonstrate the robot's capability to perform complex motions during locomotion, Fig. 8 in the appendix provides a sim-to-real comparison of the robot completing a full motion sequence while moving forward and recovering to a standing position. These experiments confirm that ALMI enables the real robot to achieve robust locomotion and accurate motion tracking.

## 7 Conclusion

This paper presented ALMI, an adversarial training framework for humanoid whole-body control. ALMI leverages the designed curriculum for upper and lower body training, achieving stable locomotion and precise motion imitation through iterative policy updates. Our theoretical analysis demonstrates ALMI's effectiveness within a two-player Markov game framework, supported by practical algorithms for implementation. The learned policy is used to collect the ALMI-X dataset with language annotations, facilitating foundation model training for the research community. Empirical results highlight ALMI's superior performance in both simulated environments and real-world deployments. However, our approach still has limitations. The adversarial framework of ALMI, albeit practical and robust for mobile manipulation, performs suboptimally in highly dynamic whole-body tasks like dancing; furthermore, our foundation model remains exploratory, with significant headroom for enhancing data efficiency. Future works include enhancing whole-body coordination through unified task-oriented rewards and improving the foundation model using advanced model architectures.

## Acknowledgments

This work is supported by the National Key Research and Development Program of China (Grant No.2024YFE0210900), the National Natural Science Foundation of China (Grant No.62306242), the Young Elite Scientists Sponsorship Program by CAST (Grant No. 2024QNRC001), and the Yangfan Project of the Shanghai (Grant No.23YF11462200).

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

# A   Theoretical Analysis

In this section, we provide more theoretical analysis for the Markov game in our method. We give the analysis of the zero-sum game to learn the locomotion policy $\pi^l$, and a similar analysis can be derived for learning $\pi^u$.

Recall that in learning $\pi^l$, we consider the lower body as *agent*, and the upper body is *adversary* that causes adversarial disturbances to the locomotion policy. Thus, the lower-body policy receives the command-following reward as $r^l(s, a^l, a^u)$, and the upper-body policy obtains a negative reward $-r^l(s, a^l, a^u)$. Formally, the value function $V^l(\pi^l, \pi^u)$ is defined as

$$V^l(s, \pi^l, \pi^u) := \mathbb{E}_{\pi^l, \pi^u} \left[ \sum_{t=0}^{T} r^l(s_t, a^l, a^u) \big| s_0 = s \right], \tag{8}$$

Then we have the value function as $V_\rho^l(\pi^l, \pi^u) = \mathbb{E}_{s \sim \rho}[V^l(s, \pi^l, \pi^u)]$, which is defined by the expectation of accumulated locomotion reward $r^l$. According to the theory of stochastic game [65], for any game $\mathcal{G}$, there exists a Nash equilibrium $(\pi_1^\star, \pi_2^\star)$ such that $V_\rho(\pi_1^\star, \pi_2) \leq V_\rho(\pi_1^\star, \pi_2^\star) \leq V_\rho(\pi_1, \pi_2^\star)$. Then we have

$$V_\rho(\pi^l, \pi^{u\star}) \leq V_\rho(\pi^{l\star}, \pi^{u\star}) \leq V_\rho(\pi^{l\star}, \pi^u), \quad \text{for all } \pi^l, \pi^u, \tag{9}$$

and in particular

$$V_\rho^{l\star} := V_\rho^l(\pi^{l\star}, \pi^{u\star}) = \max_{\pi^l} \min_{\pi^u} V_\rho^l(\pi^l, \pi^u) = \min_{\pi^u} \max_{\pi^l} V_\rho^l(\pi^l, \pi^u), \tag{10}$$

which signifies that $\pi^l$ is learned to *maximize* the value function to better follow commands in locomotion, while $\pi^u$ tries to *minimize* the value function, aiming to provide an effective disturbance to help learn a robust locomotion policy. Our goal in this setting is to develop algorithms to find $\varepsilon$-approximate Nash equilibrium, i.e. to find $\pi^l$ such that

$$\left| \min_{\pi^u} V_\rho(\pi^l, \pi^u) - V_\rho^l(\pi^{l\star}, \pi^{u\star}) \right| \leq \varepsilon, \tag{11}$$

To solve this min-max problem in Eq. (10), we adopt an independent RL optimization process for both players. Specifically, the *agent* obtains $[(s_0, a_0^l, r_0^l), \ldots, (s_T, a_T^l, r_T^l)]$, and *adversarial* obtains $[(s_0, a_0^u, r_0^u), \ldots, (s_T, a_T^u, r_T^u)]$ by executing each policy in the game to sample a trajectory, where each player is oblivious to the actions of the other player. In our analysis, we adopt continuous policy parameterizations $x \mapsto \pi^u$, and $y \mapsto \pi^l$, where $x \in \mathcal{X} \subseteq \mathbb{R}^{d_1}$, $y \in \mathcal{Y} \subseteq \mathbb{R}^{d_2}$ are parameter vectors. Each player simply treats $V_\rho^l(x, y) \mapsto V_\rho^l(\pi^u, \pi^l)$ as a continuous optimization objective, and updates their policy using REINFORCE gradient estimator [66]. For episode $i$, the two players update their policies with stochastic gradient descent, as

$$x^{(i+1)} \leftarrow \mathcal{P}_{\mathcal{X}}(x^{(i)} - \eta_x \widehat{\nabla}_x^{(i)}), \quad y^{(i+1)} \leftarrow \mathcal{P}_{\mathcal{Y}}(y^{(i)} + \eta_y \widehat{\nabla}_y^{(i)}), \tag{12}$$

where $\mathcal{P}_{\mathcal{X}}$ denotes euclidean projection onto the convex set $\mathcal{X}$, with

$$\widehat{\nabla}_x^{(i)} \mapsto R_T^{(i)} \sum_{t=0}^{T} \nabla \log \pi^u(a_t^{u(i)} \mid s_t^{(i)}), \quad \widehat{\nabla}_y^{(i)} \mapsto R_T^{(i)} \sum_{t=0}^{T} \nabla \log \pi^l(a_t^{l(i)} \mid s_t^{(i)}), \tag{13}$$

where $R_T^{(i)} \mapsto \sum_{t=0}^{T} r_t^{l(i)}$. This process is independent for two players, as they optimize policies independently with policy gradients using their own experiences. Here, it is well-known that if players update their policies independently using online gradient descent/ascent with the same learning rate, the resulting dynamics may cycle, leading to poor guarantees [67, 68]. However, previous studies show that two-timescale rules help policy converge in simple minimax optimization settings [69, 70]. As a result, we follow recent studies in the Min-Max game [71] and use *two-timescale updates* for the two players, which is a simple modification of the usual gradient descent-ascent scheme for Min-Max optimization, in which the Min player uses a much smaller step size than the Max player. Next, to ensure that the variance of the REINFORCE estimator remains bounded, we require that both players use $\varepsilon$-greedy exploration in conjunction with the basic policy gradient updates.

**Assumption A.1.** Both players follow direct parameterization with $\varepsilon$-greedy exploration, as $\pi^u(a^u \mid s) = (1 - \varepsilon_x)\mathbb{1}_{s,a^u} + \varepsilon_x/|\mathcal{A}^u|$ and $\pi^l(a^l \mid s) = (1 - \varepsilon_y)\mathbb{1}_{s,a^l} + \varepsilon_y/|\mathcal{A}^l|$, where $\varepsilon_x, \varepsilon_y \in [0, 1]$ are the *exploration factors*.

Then the following theorem holds by following Assumption A.1 and the two-timescale rule of update.

**Theorem A.2** (Restate of Theorem 3.1). *Given $\epsilon > 0$, suppose each policy has $\varepsilon$-greedy exploration scheme with factors of $\varepsilon_x \asymp \epsilon$ and $\varepsilon_x \asymp \epsilon^2$, under a specific two-timescale rule of the two players' learning-rate following the independent policy gradient, we have*

$$\max_{\pi^l} \min_{\pi^u} V_\rho(\pi^l, \pi^u) - \mathbb{E}\left[ \frac{1}{N} \sum_{i=1}^{N} \min_{\pi^u} V_\rho(\pi^u, \pi^{l(i)}) \right] \leq \epsilon \tag{14}$$

*after $N$ episodes, which results in a $\epsilon$-approximate Nash equilibrium.*

*Proof.* This proof basically follows Appendix A.2 of Daskalakis et al. [71], which proves that the Max player leads to the $\epsilon$-approximate Nash equilibrium. We can follow a similar process to prove that the Min player also has the same property. According to [71], by following the REINFORCE gradient estimator, we have

$$\mathbb{E}_{\pi^u, \pi^l} \|\widehat{\nabla}_x - \nabla_x V_\rho^l(x,y)\|^2 \leq 24 \frac{|\mathcal{A}^u|^2}{\varepsilon_x \zeta^4}, \quad \text{and} \quad \mathbb{E}_{\pi_x, \pi_y} \|\widehat{\nabla}_y - \nabla_y V_\rho^l(x,y)\|^2 \leq 24 \frac{|\mathcal{A}^l|^2}{\varepsilon_y \zeta^4}, \quad (15)$$

where $\zeta$ is the stopping probability of the Markov game. Then according to the performance difference lemma [72], we have

$$V_\rho(\pi^u, \pi^l) - V_\rho(\pi^{u'}, \pi^l) = \sum_{s \in \mathcal{S}} \tilde{d}_\rho^{\pi^u, \pi^l}(s) \mathbb{E}_{a \sim \pi^u(\cdot|s)} \mathbb{E}_{b \sim \pi^l(\cdot|s)} \left[ A^{\pi^{u'}, \pi^l}(s,a,b) \right] \quad (16)$$

Then, for a policy $\pi$, let $\pi_1^\star(\pi^l) \in \Pi_1^\star(\pi^l)$ denote a policy minimizing $\|\frac{d_\rho^{\pi_1, \pi^l}}{\rho}\|_\infty$, then we have

$$V_\rho^l(x,y) - \min_{x'} V_\rho(x', y) \leq V_\rho(\pi^u, \pi^l) - V_\rho(\pi_1^*(\pi^l), \pi^l) \quad (17)$$

$$= \sum_{s,a} \tilde{d}_\rho^{\pi_1^\star(\pi^l), \pi^l}(s) \pi_1^\star(\pi^l)(a \mid s) \mathbb{E}_{b \sim \pi^l(\cdot|s)} [-A^{\pi^u, \pi^l}(s,a,b)] \quad (18)$$

$$\leq \sum_s \tilde{d}_\rho^{\pi_1^\star(\pi^l), \pi^l}(s) \max_a \mathbb{E}_{b \sim \pi^l(\cdot|s)} [-A^{\pi^u, \pi^l}(s,a,b)] \quad (19)$$

$$\leq \left\| \frac{\tilde{d}_\rho^{\pi_1^\star(\pi^l), \pi^l}}{\tilde{d}_\rho^{\pi^u, \pi^l}(s)} \right\|_\infty \sum_s \tilde{d}_\rho^{\pi^u, \pi^l}(s) \max_a \mathbb{E}_{b \sim \pi^l(\cdot|s)} [-A^{\pi^u, \pi^l}(s,a,b)]. \quad (20)$$

We observe that $\left\| \frac{\tilde{d}_\rho^{\pi_1^\star(\pi^l), \pi^l}}{\tilde{d}_\rho^{\pi^u, \pi^l}} \right\|_\infty \leq \frac{1}{\zeta} \left\| \frac{d_\rho^{\pi_1^\star(\pi^l), \pi^l}}{\rho} \right\|_\infty \leq \frac{1}{\zeta} C_\mathcal{G}$, where $C_\mathcal{G}$ is the minimax mismatch coefficient for the game $\mathcal{G}$. Then we have

$$\sum_{s,a} \tilde{d}_\rho^{\pi^u, \pi^l}(s) \max_a \mathbb{E}_{b \sim \pi^l(\cdot|s)} [-A^{\pi^u, \pi^l}(s,a,b)]$$

$$= \max_{\bar{x} \in \Delta(\mathcal{A}^u)^{|\mathcal{S}|}} \sum_{s,a} \tilde{d}_\rho^{\pi^u, \pi^l}(s) \bar{x}_{s,a} \mathbb{E}_{b \sim \pi^l(\cdot|s)} [-A^{\pi^u, \pi^l}(s,a,b)]$$

$$= \max_{\bar{x} \in \Delta(\mathcal{A}^u)^{|\mathcal{S}|}} | \sum_{s,a} \tilde{d}_\rho^{\pi^u, \pi^l}(s)(\pi^u(a \mid s) - \bar{x}_{s,a}) \mathbb{E}_{b \sim \pi^l(\cdot|s)} [Q^{\pi^u, \pi^l}(s,a,b)]$$

$$= \max_{\bar{x} \in \Delta(\mathcal{A}^u)^{|\mathcal{S}|}} \sum_{s,a} \tilde{d}_\rho^{\pi^u, \pi^l}(s)((1-\varepsilon_x)x_{s,a} + \varepsilon_x A^{-1} - \bar{x}_{s,a}) \mathbb{E}_{b \sim \pi^l(\cdot|s)} [Q^{\pi^u, \pi^l}(s,a,b)],$$

$$\leq \max_{\bar{x} \in \Delta(\mathcal{A}^u)^{|\mathcal{S}|}} \sum_{s,a} \tilde{d}_\rho^{\pi^u, \pi^l}(s)((1-\varepsilon_x)x_{s,a} + \varepsilon_x A^{-1} - \varepsilon_x \bar{x}_{s,a} - (1-\varepsilon_x)\bar{x}_{s,a}) \mathbb{E}_{b \sim \pi^l(\cdot|s)} [Q^{\pi^u, \pi^l}(s,a,b)]$$

$$\leq (1-\varepsilon_x) \max_{\bar{x} \in \Delta(\mathcal{A}^u)^{|\mathcal{S}|}} \sum_{s,a} \tilde{d}_\rho^{\pi^u, \pi^l}(s)(x_{s,a} - \bar{x}_{s,a}) \mathbb{E}_{b \sim \pi^l(\cdot|s)} [Q^{\pi^u, \pi^l}(s,a,b)] + \frac{2\varepsilon_x}{\zeta^2}$$

$$= \max_{\bar{x} \in \Delta(\mathcal{A}^u)^{|\mathcal{S}|}} \langle \nabla_x V_\rho(x,y), x - \bar{x} \rangle + \frac{2\varepsilon_x}{\zeta^2}, \quad (21)$$

where the last equation holds since $Pr[T \geq t] \leq (1-\zeta)^t$ for any $t \geq 0$ that for any $\rho \in \Delta(\mathcal{S})$,

$$\nabla_x V_\rho(x,y) = \mathbb{E}_{\tau \sim Pr^{\pi_x, \pi_y}(\cdot|s_0)} \left[ \sum_{t=0}^T (\nabla_x \log \pi_x(a_t|s_t)) Q^{\pi_x, \pi_y}(s_t, a_t, b_t) \right] \quad (22)$$

$$= \sum_{s \in \mathcal{S}} \mathbb{E}_{a \sim \pi_x(\cdot|s)} \mathbb{E}_{b \sim \pi_y(\cdot|s)} \left[ \tilde{d}_\rho^{\pi_x, \pi_y}(s)(\nabla_x \log \pi_x(a|s)) Q^{\pi_x, \pi_y}(s,a,b) \right]. \quad (23)$$

Thus, for any $s \in \mathcal{S}, a \in \mathcal{A}$, we have

$$\frac{\partial V_\rho(x,y)}{\partial x_{s,a}} = (1-\varepsilon_x) \tilde{d}_\rho^{\pi_x, \pi_y}(s) \mathbb{E}_{b \sim \pi_y(\cdot|s)} \left[ Q^{\pi_x, \pi_y}(s,a,b) \right],$$

and so it follows that

$$\left| \frac{\partial V_\rho(x,y)}{\partial x_{s,a}} \right| \leq \frac{d_\rho^{\pi_x, \pi_y}(s)}{\zeta} \left| \mathbb{E}_{b \sim \pi_y(\cdot|s)} \left[ Q^{\pi_x, \pi_y}(s,a,b) \right] \right|.$$

According to Eq. (17) and (21), we have

$$V_\rho(\pi^u, \pi^l) - \min_{x'} V_\rho(x', \pi^l) \leq \min_{\pi_1 \in \Pi_1^*(\pi^l)} \left\| \frac{d_\rho^{\pi_1, \pi^l}}{\rho} \right\|_\infty \left( \frac{1}{\zeta} \max_{\bar{x} \in \Delta(\mathcal{A}^u)^{|\mathcal{S}|}} \left\langle \nabla_x V_\rho(\pi^u, \pi^l), \pi^u - \bar{x} \right\rangle + \frac{2\varepsilon_x}{\zeta^3} \right),$$
(24)

The term in the right side can be bounded by $O\left(\frac{\epsilon C_g}{\zeta}\right)$ according to the gradient dominance condition in Lemma 1a of [71]. Then, according to Theorem 2a of [71], suppose each policy has $\varepsilon$-greedy exploration scheme with factors of $\varepsilon_x \asymp \epsilon$ and $\varepsilon_x \asymp \epsilon^2$, the average performance difference can be bounded. Specifically, under a specific two-timescale rule of the two players' learning-rate, we have

$$\max_{\pi^l} \min_{\pi^u} V_\rho(\pi^l, \pi^u) - \mathbb{E}\left[ \frac{1}{N} \sum_{i=1}^N \min_{\pi^u} V_\rho(\pi^u, \pi^{l(i)}) \right] \leq O\left( \frac{\epsilon C_g}{\zeta} \right),$$
(25)

which concludes our proof. □

# B  Implementation Details

## B.1  State and Action Space

In this section, we introduce the detailed observation and action-space information of policies used in experiments. The adversarial iterations use the same state space setting. We use 21 DoF of the H1-2 robot without its wrist joints. The details are in Table 7.

Table 7: State and action space information in ALMI setting.

| State Term | Lower dim. | Upper dim. | Whole dim. |
|---|---|---|---|
| Base angular velocity | 3 | 3 | 3 |
| Base gravity | 3 | 3 | 3 |
| Commands | 3 (velocity) | 9 (motion) | 12 (velocity+motion) |
| DoF position | 21 | 21 | 21 |
| DoF velocity | 21 | 21 | 21 |
| Actions | 12 (lower) | 9 (upper) | 21 (whole) |
| Periodic phase | 2 | 2 | 2 |
| Total dim | 65 | 68 | 83 |

The policies use the PPO [11] algorithm for training, where the observation of the critic policy has 3 additional dimensions of base linear velocity compared to the state space of the actor policy, which is often called *privileged information* in robotics. For action space, the lower body policy uses 12 DoF of two leg joints, the upper body uses 9 DoF of one waist joint and two arms, and the whole body policy uses all 21 DoF joints.

## B.2  Training Detail of the Locomotion Policy

**Algorithm Description**    We employ PPO to train the locomotion policy, which consists of two key components: the policy and the environment. Trajectories are generated by deploying the policy on the robot within the environment, and the collected data is then used to update the policy. Additionally, the curriculum factors are used to adjust at the end of each episode. The algorithmic description is given in Algorithm 1.

**Implementation Details**    To enable the robot to exhibit a regular gait during movement and remain stationary when the velocity command is zero, we design a gait phase parameter $\phi_t = (\phi_{t,\text{left}}, \phi_{t,\text{right}})$ as an observation term, which is updated as:

$$\phi_{t+1,\text{left}} = \phi_{t,\text{left}} + f \times \mathrm{d}t,$$
(26)
$$\phi_{t+1,\text{right}} = \phi_{t+1,\text{left}} + \psi,$$
(27)

where $f, \mathrm{d}t, \psi$ are gait frequency, time step and gait offset, respectively. We use PPO to train the policy and employ an asymmetric actor-critic architecture, where the critic is granted access to the base linear velocity $v_t$ as privileged information. For the first iteration, we apply an open-loop controller to control the upper-body to track the reference motion trajectory. In subsequent iterations, we employ the trained upper-body policy as the upper-body controller.

**Reward Design**    We divide the reward terms into three parts according to their role: *penalty* rewards to prevent unwanted behaviors, *regularization* to refine motion, and *task* reward to achieve goal tracking (velocity commands or upper body DoF position). The details are in Table 1. The three iterations use the same reward terms and weights.

---

**Algorithm 1** Training process of the locomotion policy

---

**Require:** max iterations $N$, max episode length $l_{\max}^{\mathrm{sl}}$
1: Initialize policy $\pi_\theta^l$, value function $V_\phi^l$
2: **for** iteration $= 1, \ldots, N$ **do**
3:     Collect trajectories by running policy $\pi_\theta^l$ in the environment
4:     // Update curriculum at episode termination
5:     **if** $l^{\mathrm{msl}} > 0.8 \times l_{\max}^{\mathrm{sl}}$ **then**
6:         $\alpha_d \leftarrow \min(\alpha_d + w, \|\mathcal{M}\| - w)$
7:     **else**
8:         $\alpha_d \leftarrow \max(\alpha_d - 2w, 0)$
9:     **end if**
10:     **if** $\alpha_d == \|\mathcal{M}\| - w$ **then**
11:         $\alpha_s \leftarrow \min(\alpha_s + 0.05, 1), \alpha_d \leftarrow 0$
12:     **else if** $\alpha_d == 0$ **then**
13:         $\alpha_s \leftarrow \max(\alpha_s - 0.01, 0)$
14:     **end if**
15:     Perform PPO update with GAE advantage update
16: **end for**
17: **return** Trained policy $\pi_\theta$

---

### B.3   Training Detail of the Motion-Tracking Policy

**Algorithm Description**   During the training of the upper-body motion tracking policy, we use the pre-trained lower-body policy $\pi^l$ to execute the locomotion command and simultaneously introduce disturbance to the upper-body. We begin by sampling the locomotion velocity commands from a narrow range and compute the tracking error of the motion-tracking task. At each episode termination, if the tracking error is smaller than a threshold, the command sampling range will be extended to generate more intensive disturbance to the upper-body. We use PPO to train the motion tracking policy to simply follow the reference DoF position sampled from the dataset. We did not use the 6D pose of keypoints as an observation because the computation is based on the robot's root pose, which will differ from that of motion dataset under omnidirectional commands. In contrast, joint angles avoid such dependency and provide sufficient information for motion tracking.

**Curriculum Setting**   In learning the motion tracking policy in the upper body, the lower-body command follows a curriculum by adjusting the range of the velocity commands. Formally,

$$c_{\min}^l \leftarrow \begin{cases} \max(c_{\min}^l - 0.1, C_{\min}^l), & \exp(-0.5\|\hat{q}^{\mathrm{u}} - q^{\mathrm{u}}\|_2^2) > d^{\mathrm{u}} \\ \max(c_{\min}^l + 0.1, C_{\min}^l), & \text{otherwise} \end{cases}, \tag{28}$$

$$c_{\max}^l \leftarrow \begin{cases} \min(c_{\max}^l + 0.1, C_{\max}^l), & \exp(-0.5\|\hat{q}^{\mathrm{u}} - q^{\mathrm{u}}\|_2^2) > d^{\mathrm{u}} \\ \min(c_{\max}^l - 0.1, C_{\max}^l), & \text{otherwise} \end{cases}, \tag{29}$$

where $c_{\min}^l, c_{\max}^l$ are the lower and upper bound of the current command range, $c^l$ is sampled from $[c_{\min}^l, c_{\max}^l]$ during training. $[C_{\min}^l, C_{\max}^l]$ is the range limit of pre-defined commands. $\hat{q}^{\mathrm{u}}, q^{\mathrm{u}}$ and $d^u$ are the reference upper body joint position, upper body joint position and the threshold of motion tracking error, respectively. The related terms and values are given in Table 8.

Table 8: Upper body curriculum terms and values.

| Term | Value |
|------|-------|
| $C_{\min}^l$ | [-0.7, -0.5, -0.5] |
| $C_{\max}^l$ | [0.7, 0.5, 0.5] |
| $d^u$ | 0.9 |

**Reward Design**   Table 2 gives the reward terms for the motion tracking policy in the upper body. The upper-body policies of adversarial iterations all use these reward terms and weights.

### B.4   PPO Hyperparameters

We use GAE to estimate the advantage and CLIP-PPO to train our policy. The total loss is given by:

$$L_{\mathrm{total}} = L_{\mathrm{policy}} + w_v L_{\mathrm{value}} - w_s S, \tag{30}$$

where

$$L_{\text{policy}} = -\mathbb{E}[\min(r_i(\theta)\hat{A}_i, \text{clip}(r_i(\theta), 1 - \epsilon, 1 + \epsilon)\hat{A}_i)], \quad (31)$$

$$L_{\text{value}} = \mathbb{E}[(V(s_i) - R_i)^2], \quad (32)$$

$$S = \mathbb{E}[-\pi_\theta(a|s)\log \pi_\theta(a|s)], \quad (33)$$

where $r_i(\theta) = \frac{\pi_\theta(a_i|s_i)}{\pi_{\theta_{old}}(a_i|s_i)}$ and $\hat{A}_i$ is the estimated advantage. $w_v$ and $w_s$ are value loss coefficient and entropy term coefficient, respectively. The hyperparameters of the PPO algorithm and the information of the network backbone are listed in Table 10.

## B.5 Domain Randomization

Domain randomization is a popular technique for improving domain transfer, often used in a zero-shot setting when the target domain is unknown or cannot easily be used for training [73]. In order to adapt the trained policy to the real world, we employ the domain randomization technique during training to facilitate robust sim-to-sim and sim-to-real transfer [36, 37]. We give the domain randomization terms and ranges used during the training process, and the details are in Table 9.

Table 9: Domain randomization terms and ranges.

| Term | Value |
|---|---|
| **Dynamics Randomization** | |
| Friction | $\mathcal{U}(0.1, 1.25)$ |
| Base mass | $\mathcal{U}(-3, 5)$ kg |
| Link mass | $\mathcal{U}(0.9, 1.1) \times$ default kg |
| Base CoM | $\mathcal{U}(-0.1, 0.1)$ m |
| Control delay | $\mathcal{U}(0, 40)$ ms |
| **External Perturbation** | |
| Push robot | interval = 10s, $v_{\text{xy}} = 1$m/s |
| **Randomized Terrain** | |
| Terrain type | trimesh, level from 0 to 10 |
| **Velocity Command** | |
| Linear x velocity | $\mathcal{U}(-1.0, 1.0)$ |
| Linear y velocity | $\mathcal{U}(-0.3, 0.3)$ |
| Angular yaw velocity | $\mathcal{U}(-0.5, 0.5)$ |

Table 10: Hyperparameters related to PPO

| Hyperparameter | Default Value |
|---|---|
| Actor lstm size | [64] |
| Actor MLP size | [64, 32] |
| Critic MLP size | [64, 32] |
| Optimizer | Adam |
| Batch size | 4096 |
| Mini Batches | 4 |
| Learning epoches | 8 |
| Activation | elu |
| Entropy coef($w_s$) | 0.01 |
| Value loss coef($w_v$) | 1.0 |
| Clip param | 0.2 |
| Max grad norm | 1.0 |
| Init noise std | 0.8 |
| Learning rate | 1e-3 |
| Desired KL | 0.01 |
| GAE decay factor($\lambda$) | 0.95 |
| GAE discount factor($\gamma$) | 0.998 |
| Curriculum window size($w$) | 40 |

## C Data Collection and Model Details

**Dataset Overview** The ALMI-X dataset contains 1989 motions combined with 41 commands, resulting 81,549 trajectories totally. Each trajectory is organized as $\tau = \{\tau_s^t, \tau_a^t, \tau_{\text{dof\_pos}}^t, \tau_{\text{trans}}^t, \tau_{\text{rot}}^t\}_{t=1}^T$, respectively representing robot states, actions, joint angles, global position and global orientation, where the global position and global orientation are usually considered as privilege information that is missing in the state space. The statistical results shown in Fig. 4, Fig. 5 and Fig. 6 demonstrate the advantages of ALMI-X, and the experimental results of Appendix D further indicate that our dataset is higher quality and more suitable for foundation model training. The dataset is available at `https://almi-humanoid.github.io/`.

**Data Collection Details** (i) For the lower-body, we first categorize command directions into 11 types (8 translation directions: front, back, left, right, front-left, front-right, back-left, and back-right; 2 rotation directions: left and right; and keep standing) according to different combination of linear and angular velocity command, and define 4 difficulty levels for command magnitudes, as shown in Table 11. We sample velocity value for the first 3 levels and task a fix value for the 'fix' level, which corresponds

Table 11: Command difficulty level setting.

| | **Command** | | |
|---|---|---|---|
| **Level** | $\hat{v}_{\text{x,t}}$ | $\hat{v}_{\text{y,t}}$ | $\hat{\omega}_{\text{yaw,t}}$ |
| easy | [0.2,0.4] | [0.2,0.3] | [0.2,0.3] |
| medium | [0.4,0.5] | [0.3,0.4] | [0.3,0.4] |
| hard | [0.5,0.7] | [0.4,0.5] | [0.4,0.5] |
| fix | 0.4 | 0.4 | 0.4 |

to a text that does not contain speed-related modifiers, e.g. just 'go forward' without 'slowly' or 'fast'. We take the lower-body commands according to the difficulty levels with corresponding ranges, combining with different direction types. In addition, we set a special category for standing, resulting in 41 categories totally. (ii) For the upper body, we select motions according to §4 and remove those that are overly large in amplitude or difficult

to distinguish, resulting in 680 different motions. However, these motions have an uneven distribution, with each motion category associated with a different number of trajectories. As shown in the left plot of Fig. 4, for instance, the number of steps for "wave" greatly exceeds that of "salute". Through empirical evaluation, we find that imbalanced data distribution negatively impacts the training performance of the foundation model.

To address this problem, we classify the motions into 30 categories based on their corresponding texts. Then we count the number of steps in each category and augment those with fewer steps by repeating their motion sequences multiple times during data collection. This is not equivalent to mere data duplication, since environmental interactions cause the robot to produce slightly different trajectories each time, despite following the same reference motion. After that, the data volume across all categories is approximately the same, as shown in Fig. 4.

After data augmentation, we obtain a total of 1,989 motions. When combined with different direction and difficulty commands, these motions generate a total of 81,549 trajectory sequences. Fig. 5 shows the planar position of all steps in space. The visualization result shows that despite the accumulated errors from the policy and the limitations of the simulator, the robot is still able to execute velocity commands in a reasonably correct manner. For instance, under the 'go forward' command, the steps are distributed along the positive x-axis, enabling the foundation model to learn representations across the space.

Fig. 6 illustrates the distribution of the upper body-hand positions of dataset. The visualization shows that the upper body motions in the dataset basically allow the robot arms to cover the entire activity space. When we train the lower body policy, we set the arm curriculum which is detailed in §3.2 to gradually expand the activity range of the two arms from the default position to the whole activity space.

**Comparison to Humanoid-X [13] data**  Compared to the existing text-to-action dataset, Humanoid-X [13], our dataset exhibits much higher quality, which can be utilized for relatively stable open-loop control of the robot in simulations without causing the robot to lose control. This is because the RL policy that UH-1 employs to track motions have complex sources and can introduce dangerous or unreasonable behavior. However, due to the robust lower-body policy and expressive upper-body policy obtained through adversarial and curriculum learning, our collected ALMI-X dataset exhibits high executability. This attribute benefits the learning of the robot foundation model via a supervised learning paradigm, since the learned model would be better for deployment and also ensures safety in the real robot.

# D  Experiment Results of Foundation Model

In this section, we introduce the architecture details and inference process of the Transformer-based foundation model. Then, we give experimental results and analysis of the trained model.

**Architecture Details**  During the training process, we first extract a segment of states and actions with fixed length $H$ and its corresponding language description $\mathcal{T}$, then concatenate the state and action for the same frame resulting in a new sequence $\{(s_i, a_i), (s_{i+1}, a_{i+1}), ..., (s_{i+H-1}, a_{i+H-1})\}$, each of which is projected onto an embedding vector $x_i = \texttt{Linear}(o_i, a_i)$. We use CLIP [74] and a linear layer process the language description $\mathcal{T}$ into a vector of the same dimension $x_{\text{text}} = \texttt{Linear}(\texttt{CLIP}(\mathcal{T}))$ These vectors serve as input to the transformer decoder, which predicts the subsequent actions corresponding to each input. The computation process of the $l$-th transformer decoder layer and the output layer can be represented as follows:

$$x_{\text{text}}^l, x_i^l, ..., x_{i+H-1}^l = \texttt{CausalSA}(x_{\text{text}}^{l-1}, x_i^{l-1}, ..., x_{i+H-1}^{l-1}), \tag{34}$$

$$\{\hat{a}_{i+1}, ..., \hat{a}_{i+H}\} = \texttt{Linear}(x_{\text{text}}^N, x_i^N, ..., x_{i+H-1}^N) \tag{35}$$

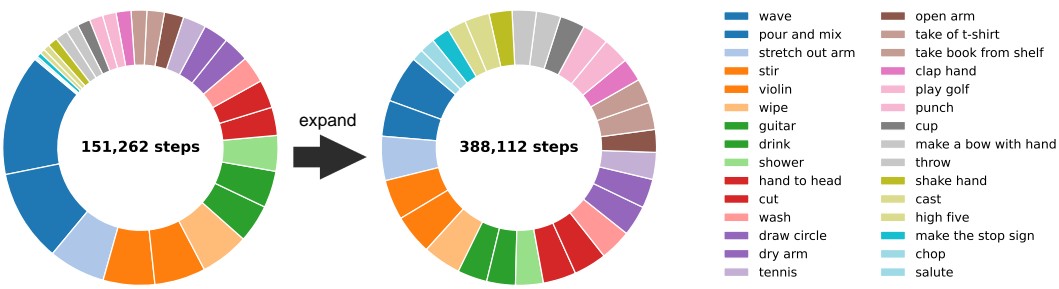

Figure 4: Percentage of steps for different categories of motions before and after data augmentation.

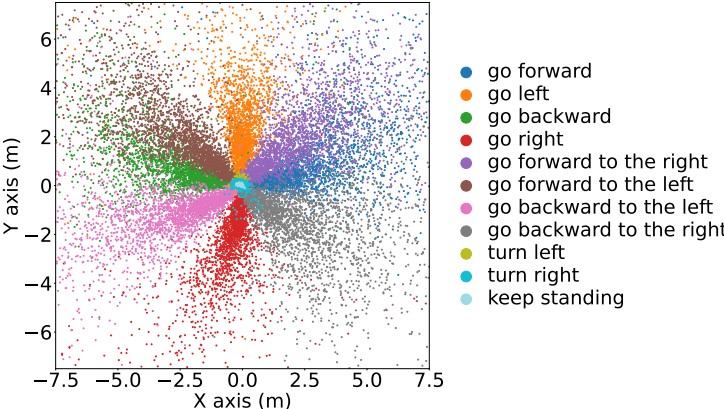

Figure 5: The visualization of $x - y$ coordinates of the robot for each step in the dataset. We down-sample the data for visualization.

where $N$ is the number of transformer decoder layers. $\hat{a}_j$, where $i + 1 \leq j \leq i + H + 1$, represent the predict action integrating text command information and historical state-action pairs. Mean square errors between predict and ground truth actions are used for policy update.

**Model Inference**    In inference, the input text vector $t_{\text{text}}$ remains constant throughout the execution of the task. The robot executes the action decoded from the last vectors of the policy output. It is then concatenated with the newly acquired state, which is then computed into an embedding vector and serve as the last vector of the input for the policy to predict the next action. In the initial stage, when the length $h$ of the historical state-action pairs is less than $H$, the robot will execute the action decoded using the vector corresponding to the latest state-action pair rather than the last one.

**Implementation Details**    We investigated multiple designs for language-guided whole-body humanoid control through supervised learning using the ALMI-X dataset. We evaluate the Transformer architecture with different input sequence lengths for the *closed-loop* robot control approach, while also exploring the *open-loop* control approach of prediction and execution of the entire sequence with a tokenizer like UH-1 [13]. Specifically, the Transformer is trained with different state-action sequence lengths. We selected 20 and 400 as the input sequence lengths for the *closed-loop* control, while the *open-loop* control approach predicts the entire motion sequence with a maximum sequence length of 700 (175 after tokenization). For a shorter sequence, we expect the model to focus more on short-term historical information, allowing more robust locomotion performance. As for the longer sequence, we aim to integrate information from complete motion sequences in modeling, thereby better responding to text commands. For tokenization, we employ the same VQ-VAE [75] architecture as in

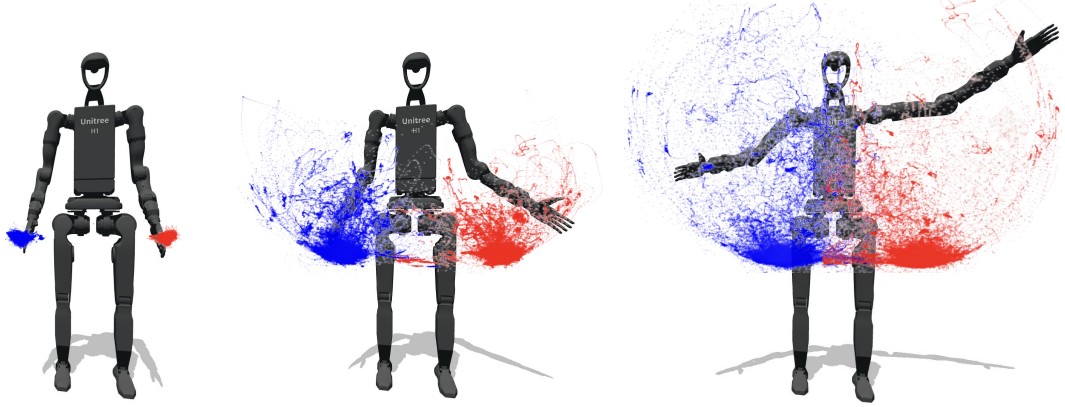

Figure 6: We illustrate the hand positions in the dataset relative to the robot's hand joints. From left to right, scaling factors of 0.05, 0.5, and 1.0 are applied to the arm joint angle offsets from the default pose, effectively modulating the motion amplitude. Larger scaling results in more pronounced arm movements.

Table 12: Average velocity of Foundation models with different text commands.

| Text commands w.r.t. lower body | Average Velocity | | |
|---|---|---|---|
| | $\bar{v}_x$ | $\bar{v}_y$ | $\bar{\omega}_{\text{yaw}}$ |
| CL-20sl | | | |
| forward fast | 0.48 | 0.18 | 0.08 |
| left fast | 0.07 | 0.28 | 0.00 |
| backward to left slowly | -0.19 | 0.21 | 0.00 |
| turn left slowly | 0.02 | 0.02 | 0.08 |
| go right fast | -0.11 | -0.37 | -0.07 |
| keep standing | 0.00 | 0.00 | 0.01 |
| CL-400sl | | | |
| forward fast | 0.87 | -0.05 | -0.10 |
| left fast | 0.53 | 0.42 | -0.07 |
| backward to left slowly | 0.35 | 0.30 | -0.01 |
| turn left slowly | 0.53 | 0.16 | 0.2 |
| go right fast | 0.47 | -0.50 | -0.10 |
| keep standing | 0.22 | 0.08 | 0.00 |

Table 13: Survival duration ($SD$) and success rates of upper-body($SR_{\text{up}}$)/lower-body($SR_{\text{low}}$) with different text commands.

| Text commands | Metrics | | |
|---|---|---|---|
| | $SR_{low}$ | $SR_{up}$ | $SD$ |
| CL-20sl | | | |
| go forward slowly and wave left. | 1.00 | 0.20 | 8.0 |
| go backward moderately and wave right. | 1.00 | 0.20 | 8.0 |
| go right fast and wave both. | 1.00 | 0.40 | 8.0 |
| CL-400sl | | | |
| go forward slowly and wave left. | 1.00 | 1.00 | 2.14 |
| go backward moderately and wave right. | 1.00 | 1.00 | 2.54 |
| go right fast and wave both. | 0.00 | 1.00 | 3.57 |
| OL | | | |
| go forward slowly and wave left. | 0.00 | 1.00 | 0.31 |
| go backward moderately and wave right. | 0.00 | 1.00 | 0.29 |
| go right fast and wave both. | 0.00 | 1.00 | 0.32 |

T2M-GPT [76] to tokenize four state-action pairs into one token; while for the non-tokenized design, we utilize a linear layer to project both the text command feature and state-action pairs into a shared latent space.

**Evaluation Metrics** Since training a foundation model to follow all text commands in ALMI-X is quite challenging, we conduct preliminary investigations using only wave-relative motion. We evaluate the models using different control approaches and input sequence lengths with different text commands in MuJoCo simulations, repeating each command 5 times with each test lasting 8 seconds. we adopt the following notation: $\text{CL} - x\text{sl}$ denotes the closed-loop control approach without tokenization, where $x$ represents the input sequence length (e.g., $\text{CL} - 20\text{sl}$). OL indicates the open-loop control approach with tokenization, which predicts and executes the entire motion sequence in a single forward pass. We evaluated several metrics, including robot survival time (up to 8 seconds), velocity during test time, and success rate of lower-body and upper-body. We consider the lower-body successful if the robot's movement direction matches the text command, and the upper-body successful if the robot waves the correct hand.

**Experiment Results** Results in Table 13 demonstrate that the open-loop control approach, which predicts and executes the entire action sequence based on a single text command, fails to maintain the robot's balance

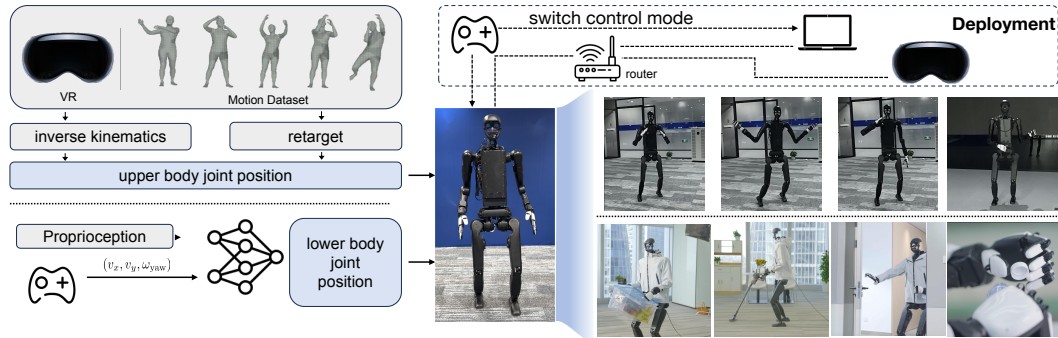

Figure 7: Real word deployment overview. The deployment framework contains an Apple Vision Pro, a router, a Unitree H1-2 robot and a PC which runs our policy and teleoperation server. The upper-body target DoF pos is given by teleoperation or motions from our motion dataset. We can switch the control mode via the remote controller. Through this framework, the robot can perform upper-body motion during movement, and diverse loco-manipulation tasks.

and leads to an extremely low survival time. In contrast, the closed-loop control method exhibits more robust locomotion and superior lower-body text-following performance. Table 12 demonstrates that the closed-loop approach with a short input sequence length enables robot's movement direction adapts to commands such as $forward$ and $right$ and its velocity adjusts to modifiers like $fast$ and $slowly$, aligning more accurately with the sampled velocity commands as shown in Table 11. However, the approach with a longer sequence length results in poorer lower-body motion performance in both movement direction and velocity. It can be inferred that the long input sequence length plays a crucial role in the upper-body success rate, since OL and $CL - 400sl$ achieve much better upper-body compliance with text commands, and focusing the model exclusively on short-term historical information brings locomotion stability and better command performance of the lower body. These experimental results indicate that training a humanoid foundation control model using supervised learning and a large-scale high-quality dataset represents a feasible approach with substantial exploratory potential. However, we find that the current model faces challenges when extending the training data to the entire ALMI-X, indicating limitations of the current model in handling highly diverse behaviors.

# E   More Experiments Regarding Adversarial Training

We also conducted more experiments to verify the rationality and advantages of the simplification to the original method.

Without simplifying the max-min problem, four separate policies would be required in each round of policy learning. Specifically, in round 1, when updating, an additional adversary is required; when updating, an additional adversary is required. It would be computationally expensive to train an additional adversary in each round. As a result, we change the inner-loop optimization of the adversary from the parameter space to the command space, which allows us to sample adversarial command to approximate the effects of training adversarial policies. We add additional experiments to show the necessity of the simplification:

**Train an Adversary.**   If we do not adopt the simplified approach, we would instead directly train an adversary to attack the policy. However, it is very challenging to properly control the strength of the adversary. Take the training of lower body policy as example, we jointly train an upper-body adversary to minimize the lower-body reward. We vary $action\_clip\_scale$ to control adversary strength, where $action\_clip = 100 \times action\_clip\_scale$. The results are shown in the Table 14. It can be seen that large adversary collapse training, while weak adversaries yield poor robustness. In contrast, our curriculum-based method, guided by motion difficulty, provides effective and stable adversarial training.

Table 14: Comparison before and after simplification of the adversarial training.

| Method | Metrics | | | | | | | |
|---|---|---|---|---|---|---|---|---|
| | $E_{\mathrm{vel}} \downarrow$ | $E_{\mathrm{ang}} \downarrow$ | $E_{\mathrm{jpe}}^{\mathrm{upper}} \downarrow$ | $E_{\mathrm{kpe}}^{\mathrm{upper}} \downarrow$ | $E_{\mathrm{action}}^{\mathrm{upper}} \downarrow$ | $E_{\mathrm{action}}^{\mathrm{lower}} \downarrow$ | $E_{\mathrm{g}} \downarrow$ | Survival $\uparrow$ |
| ALMI (Ours) | **0.2202** | **0.4812** | **0.2116** | **0.0458** | **0.0600** | **0.0175** | **0.8551** | **0.9723** |
| $action\_clip\_scale = 0.1$ | 1.2610 | 5.3445 | / | / | / | 6.4459 | 1.7446 | 0 |
| $action\_clip\_scale = 0.01$ | 1.3707 | 6.8632 | / | / | / | 6.1924 | 2.3056 | 0 |

**Policy with adversarial goals.**   Another way to implement adversarial training is to assign each policy two objectives. For this experiment, we train upper and lower policies jointly, each maximizing its own reward while minimizing the other's. We vary adversarial strength via reward weights (e.g., -1 for strong adversary and -0.1 for weak adversary). Table 15 show adversarial weight greatly affects training, and tuning it is nontrivial. The training of weak $\pi_a^u +$ strong $\pi_a^l$ and strong $\pi_a^u +$ strong $\pi_a^l$ are failed. Our simplified method ensures stable and effective adversarial interaction. Among converged settings, our method outperforms others in all metrics.

Table 15: The lower and upper policies are updated simultaneously, and both have two sets of goals.

| Method | Metrics | | | | | | | |
|---|---|---|---|---|---|---|---|---|
| | $E_{\mathrm{vel}} \downarrow$ | $E_{\mathrm{ang}} \downarrow$ | $E_{\mathrm{jpe}}^{\mathrm{upper}} \downarrow$ | $E_{\mathrm{kpe}}^{\mathrm{upper}} \downarrow$ | $E_{\mathrm{action}}^{\mathrm{upper}} \downarrow$ | $E_{\mathrm{action}}^{\mathrm{lower}} \downarrow$ | $E_{\mathrm{g}} \downarrow$ | Survival $\uparrow$ |
| ALMI (Ours) | **0.2202** | **0.4812** | **0.2116** | **0.0458** | **0.0600** | **0.0175** | **0.8551** | **0.9723** |
| weak $\pi_a^u +$ weak $\pi_a^l$ | 0.3415 | 1.4413 | 0.29469 | 0.0526 | 0.0970 | 0.0183 | 1.0250 | 0.9612 |
| strong $\pi_a^u +$ weak $\pi_a^l$ | 1.1220 | 4.6715 | 2.7251 | 0.1840 | 0.2450 | 4.6937 | 1.1702 | 0.3524 |

# F   Real-Robot Deployment

**Hardware and Deployment**   We conduct experiments using the Unitree H1-2 robot equipped with the ROBOTERA XHAND robotic hand, applying ALMI-trained policies to the lower body control and various interfaces (i.e., open-loop controller, ALMI policy, and VR device) for the upper body control, as shown in

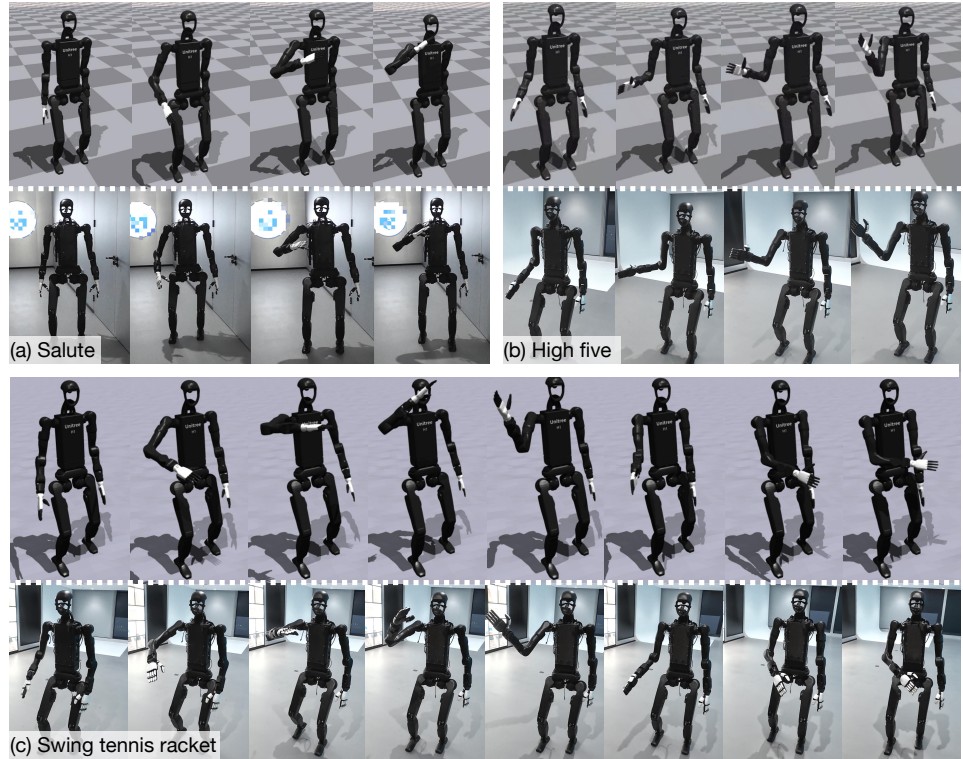

Figure 8: Upper body behavior of the robot during locomotion compared between simulator and real world. In simulator, robot keeps standing to show the original upper body motion; In real world, robot goes forward while doing motion.

Fig. 7. Specifically, (i) we can use an open-loop controller that sends the target joint positions of the pre-loaded motions; (ii) we can obtain actions of the upper body via the policy $\pi^u$, which uses the reference joint positions of the preloaded motion as input. In real-world experiments, we find that open-loop control has higher tracking accuracy when the robot remains standing, but the learned policy shows better stability during movement. (iii) The other one is combining with teleoperation systems, which enable the robot to perform loco-manipulation tasks. Specifically, the upper-body follows wrist poses of humans and the relative positions between the robot end-effectors and the robot head are expected to match those between the human wrists and head, then the arm actions are obtained from inverse kinematics. The finger movements are also captured by VR simultaneously to control the dexterous hands for loco-manipulation tasks. Benefiting from the superiority of the training method and the use of domain randomization, our policy does not require additional sim-to-real techniques and can be deployed directly on the NVIDIA Jetson Orin NX onboard the robot for inference. The action output frequency of the policy is 50Hz.

**Teleoperation for Loco-Manipulation.** Due to the absence of hand keypoints in the motion dataset, we use the VR device to capture human hand poses. This data was then used to real-time control the robot's dexterous hand, enabling it to perform various loco-manipulation tasks. In this setup, the robot's camera provides real-time, first-person 3D visual feedback to the VR device, allowing the operator to see through the robot's perspective. The system translates human wrist poses into the robot's coordinate frame, ensuring that the relative positions between the robot's end-effectors and head mirror those of the human wrists and head. The robot wrist orientations are calibrated to match the absolute orientations of the human wrists, as determined during the initialization of the hand-tracking system. We employ closed-loop inverse kinematics, implemented with Pinocchio [77], to calculate the robot arm's joint angles. The human hand keypoints are translated into robot joint angle commands through dex-retargeting, utilizing a flexible and efficient motion retargeting library [78]. Our method further leverages vector optimizers to enhance the performance of the dexterous hand control. The teleoperation system basically follows Open-Television [50].

**Real-robot Experiments on Motion Tracking** In the real world, we use the remote controller to control the robot to complete a series of motion tracking tasks. Fig. 8 provides some illustrative examples of our ALMI's motion tracking quality compared between simulation and real world. These results demonstrate that our policy

achieves comparable performance in tracking speed commands and upper-body motions in the real world as in the simulator, without relying on any additional sim-to-real techniques.

**Real-robot Experiments on Loco-Manipulation**    By leveraging the ALMI policy and teleoperation deployment, we enable the robot to accomplish a variety of real-world loco-manipulation tasks. Fig. 9 illustrates that the robot can hit the tennis ball with a precise swing. Our robot can also complete household tasks, which requires a combination of stable movement with manipulation capabilities. In Fig. 10(a), the robot can carry a 5kg box and walk stably; while in Fig. 10(b), the robot can use a vacuum to clean the floor.

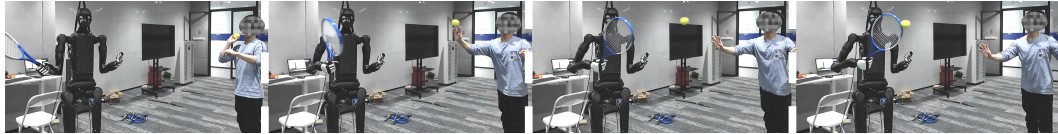

Figure 9: Robot swings the tennis racket with precision.

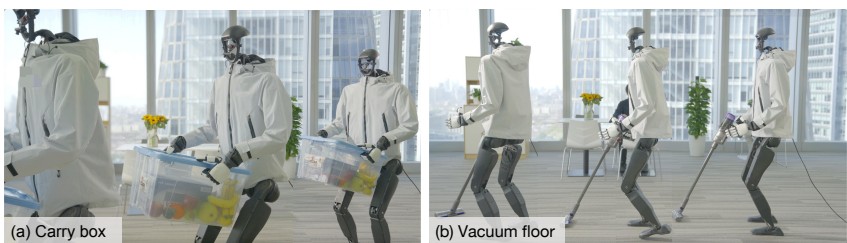

Figure 10: Robot performs loco-manipulation tasks with Teleoperation. (a) Carry a 5kg box. (b) Vaccum floor.

# G    Broader Social Impact

Our humanoid locomotion method with teleoperation enables robots to perform diverse tasks in human environments, with potential applications in disaster response, healthcare, and industrial automation. However, challenges such as job displacement and safety risks must be addressed to ensure responsible deployment. Ethical guidelines and policy discussions will be crucial to maximize societal benefits while minimizing unintended consequences.

