# OpenReview forum: "Adversarial Locomotion and Motion Imitation for Humanoid Policy Learning"
_NeurIPS.cc/2025/Conference — NeurIPS 2025 poster_

### Official Review · Reviewer_XwBe · 2025-07-03

**Clarity:** 3
**Significance:** 2
**Originality:** 3
**Rating:** 4
**Confidence:** 5

**Summary:**

Training curriculum for whole body humanoid controller has been an active area of research lately. This paper proposes ALMI (Adversarial Locomotion and Motion Imitation) . They address this by first treating lower and upper bodies separately. then by training upper and lower body contrllers separately while dilating the other policy to be adverserial to learning of the other policy.  the instantiatione implicitly assumes that lower bod precise footstep and movement is not needed and we only care about velocity. Additionally, the authors release ALMI-X, the first large-scale whole-body control dataset with 80K+ trajectories and language descriptions, enabling foundation model training for humanoid control.

**Questions:**

One of my main reason for rejection is that I am not quite sold on the notion that upper and lower body needs to be *trained separately*. I do understand that having different policy heads or networks for upper and lower body may have some benefits and should be treated differently in the reward design, but I don't see why they cannot be trained together with some clever sampling of upper and lower body command directives. I am happy to hear some counter arguments to this and am willing to change my score in rebuttal.

**Ethical Concerns:**

["NO or VERY MINOR ethics concerns only"]

**Final Justification:**

The splitting of the model and adversarial sampling of training distribution is novel enough and does not seem to come with heavy computation costs and could be complementary to other training curriculums/approaches.

**Limitations:**

Yes

**Quality:**

2

**Strengths And Weaknesses:**

*Strengths:*
- Code has been provided and is clean with enough intructions for reproducibility. The details provided in the paper is also good.
- Mujoco dataset shows sim2sim transfer and the dataset collected is good and showcases that such can be used as a dataset for training foundation model directly that can do sim2real transfer. training policy in high throughput sim, and using that to collect data in high fidelity sim to transfer to real world is a good approach.
- having adversarial curriculum is an intuitive approach to having more robust policy.

*Weakness:*
- Overall the novelty of the paper boils down to iterative adversarial upper and lower body commands curriculum. It is not clear how much effective it is as compared to other innovations already done in whole body imitation controllers like Masked Humanoid Controller, ExBody2, OmniH20, Hover, MaskedMimic to name a few. It could be wonderful to see some of these baselines for fair comparison.
- The iterative process takes longer time for training, the compute need of the approach is not clear. does all approaches in the baseline use the same amount of compute and training time, if not it has to be clear when comparing the baselines.
- There should be a way to do the iterative approach in an integrated manner with a smaller window for switching between upper and lower bod policy. having a fixed number of iteration seems a bit restrictive and compute heavy.

---

> ### Author Rebuttal · Authors · 2025-07-31
>
> Thank you for your valuable feedback and insightful comments. We appreciate the opportunity to clarify and improve our work.
>
> **W1: More baseline comparison**
>
> Thank you for the suggestion. We have added new baseline methods: OmniH2O, Exbody2 and compared them with our method on Unitree G1 robot. Due to space limits, please refer to our **response to Reviewer RhvN, W3**.
>
> **W2: Computational cost**
>
> Thank you for your insightful comment. The table below provides a detailed list of the training times and selected checkpoints for the methods being compared in the baseline experiments.
>
> ||Training Time Consumption|Experiment Checkpoint|
> |-|-|-|
> |**H1-2**|
> |ALMI (Ours)|Lower-1\~3: 10k, 6k, 5k iterations with 2.5s/iter; Upper-1\~2: 3k, 1k iterations with 2.0s/iter;  **Total: about 16.8 hours**|Lower-3 5k iter + Upper-2 1k iter|
> |ALMI (Whole)|20k iterations with 3.0s/iter; **Total: about 16.6 hours**| 20k iter|
> |ALMI (w/o curriculum)|Lower-1\~3: 15k, 8k, 6k iterations with 2.5s/iter; Upper-1\~2: 5k, 3k iterations with 2.0s/iter; **Total: about 24.5 hours**|Lower-3 6k iter + Upper-2 3k iter|
> |ALMI (Lower1)|Same as ALMI (Ours)|Lower-1 10k iter + Upper-2 3k iter|
> |ALMI (Lower2)|Same as ALMI (Ours)| Lower-2 6k iter + Upper-2 3k iter|
> |Exbody|25k iterations with 2.0s/iter; **Total: about 13.8 hours**|25k iter|
> |**G1**|
> |ALMI (ours)|Lower-1～3: 10k, 8k, 6k iterations with 2.5s/iter; Upper-1～2: 3k, 3k iterations with 2.0s/iter. **Total: about 20.0 hours**|Lower-3 6k iter + Upper-2 3k iter|
> |OmniH2O|Teacher: 30k iterations with 1.5s/iter; Student: 10k iterations with 1.5s/iter; **Total: about 16.6 hours**|Student 10k iter|
> |Exbody2|Teacher: 20k iterations with 2.5s/iter; Student: 4k iterations with 2.0s/iter; **Total: about 16.1 hours**|Student 4k iter|
>
> **W3: An integrated manner for switching policy updating**
>
> We sincerely thank the reviewer for this insightful suggestion. We agree that a fixed iteration number for alternating between upper and lower body policies can be restrictive and computationally inefficient.
>
> To address this, we designed an automatic switching mechanism. Specifically, both the upper and lower policies are initialized at the start of the training, but only one policy is updated per phase. We monitor the ratio of the current reward to its maximum, and when it surpasses a certain switching threshold (determined based on empirical observations), the PPO runner switches to update the other policy. Preliminary results are shown in the below.
>
> |Method|$E_\text{vel}\downarrow$|$E_\text{ang}\downarrow$|$E_\text{jpe}^\text{upper}\downarrow$|$E_\text{kpe}^\text{upper}\downarrow$|$E_\text{action}^\text{upper}\downarrow$| $E_\text{action}^\text{lower}\downarrow$| $E_\text{g}\downarrow$ | $\text{Survival}\uparrow$|
> |-|:-:|:-:|:-:|:-:|:-:|:-:|:-:|:-:|
> |ALMI (Ours)|**0.2202**|**0.4812**|**0.2116**|**0.0458**|0.0600| **0.0175**|**0.8551**|**0.9723**|
> |Auto Switch @10k|0.8693|1.2887|1.0391|0.0953|0.0843| 0.6747|0.9253|0.3458|
> |Auto Switch @20k|0.3677|0.9629|0.2616|0.0669|0.0674| 0.0582|0.8532|0.9389|
> |Auto Switch @30k|0.3251|0.9433|0.2563|0.0593|**0.0584**| 0.0361|0.9155|0.9402|
>
> We find this automatic switching strategy effective, as it simplifies the training workflow and reduces overall training time. However, it is worth noting that the approach is sensitive to the choice of switching criteria. If the threshold or reward metric is not well-tuned, the resulting policy may suffer from suboptimal performance. Moreover, in real-world robot control, higher rewards don't always correlate perfectly with better physical execution due to sim2real gaps.
> Overall, we believe the robust automatic switching mechanisms within adversarial training frameworks for humanoid control is an important and meaningful research avenue.
>
> **Q1: Issue about separately training upper and lower policy**
>
> We sincerely thank the reviewer for the thoughtful comments. We understand the reviewer's confusion, as some prior works (e.g., Exbody) have indeed achieved upper-body tracking and lower-body locomotion in a single training stage by assigning separate reward functions to different body parts. However, the focus of our work lies in enhancing the robustness of robot control through adversarial interactions between the upper and lower body. To this end, we introduce the theoretical framework of a two-player Markov game.
> We will first briefly explain this framework, and then use two newly added experiments to illustrate why we adopt the proposed approach in our paper.
> (i) According to our theoretical analysis, the two policies can be considered as two players in a Markov game in policy learning. Specifically, when learning the lower-body policy $\pi^l$ by maximizing it value function, the upper-body policy $\pi^u$ is optimized to minimize it and serves as an adversary. However, to solve such a min-max problem and obtain a Nash equilibrium, the update should follow a specific two-scale learning rate and independent policy gradients according to Theorem 3.1. (ii) In practice, if we use a single network to represent $\pi^l$ and $\pi^u$, the two policies are hard to obtain independent policy gradients or using two-scale learning rate, which can lead to unstable policy updates. (iii) Additionally, we **add two experiments** to further clarify our design choices:  **1)** why we simplify the adversarial process using a command-based curriculum and **2)** why the two policies are trained separately.
>
> **1) Train an adversary.**
> If we do not adopt the simplified approach, we would instead directly train an adversary to attack the policy. **However, it is very challenging to properly control the strength of the adversary**. Taking the training of lower-body policy as an example, we jointly train an upper-body adversary whose objective is to minimize the reward of the lower-body policy.
> To reflect different levels of adversarial strength, we vary the `action_clip_scale` parameter. Specifically, the upper-body's action is clipped using a scaled range:
> `actual_action_clip = 100 × action_clip_scale`. The results are shown in the table below. We observe that different settings of `action_clip_scale` significantly affect the training outcome. With `action_clip_scale=1`, the adversary tends to produce large action outputs, which quickly cause the robot to fail and eventually lead to training collapse.
> A moderate `action_clip_scale` creates strong interference for the lower-body policy—resulting in slow reward improvement—but does not entirely destabilize the policy. In contrast, a small `action_clip_scale` leads to weak adversarial influence, with negligible impact on the lower policy’s performance.
>
> This experiment demonstrates that the lower policy is highly sensitive to the strength of adversarial attacks. When the adversary and the policy are updated simultaneously, the adversary must not be too strong to prevent training collapse, yet still provide sufficient challenge to enhance the robustness of the lower policy.
> Therefore, adopting a curriculum-based adversary is an intuitive and effective solution. Our approach—using the difficulty of reference motions as the basis for the curriculum—**not only ensures coverage of common robot behaviors, but also enables progressive adversarial training**.
>
> | Upper body action clip scale  | Mean episode length @20k | Mean reward @20k|
> |:-:|:-:|:-:|
> |1|7.0|0.0|
> |0.1|612.3|4.21|
> |0.01|978.5|25.79|
>
> |Method|$E_\text{vel}\downarrow$| $E_\text{ang}\downarrow$| $E_\text{jpe}^\text{upper}\downarrow$| $E_\text{kpe}^\text{upper}\downarrow$| $E_\text{action}^\text{upper}\downarrow$| $E_\text{action}^\text{lower}\downarrow$| $E_\text{g}\downarrow$|$\text{Survival}\uparrow$|
> |-|:-:|:-:|:-:|:-:|:-:|:-:|:-:|:-:|
> |ALMI (Ours)|**0.2202**|**0.4812**|**0.2116**|**0.0458**|**0.0600**|**0.0175**|**0.8551**|**0.9723**|
> |`action_clip_scale`=0.1|1.2610|5.3445|\\|\\|\\|6.4459|1.7446|0|
> |`action_clip_scale`=0.01|1.3707|6.8632|\\|\\|\\|6.1924|2.3056|0|
>
> **2) Policy with adversarial goals.**
> Another way to implement adversarial training is to assign each policy two objectives. For this experiment, we train upper and lower policies jointly, each maximizing its own reward while minimizing the other’s.
> We vary adversarial strength via reward weights (**e.g., -1 for strong adversary and -0.1 for weak adversary**). The experimental results are shown in the table below. It can be observed that the weight of the adversarial goal also has a significant impact on training, and selecting an appropriate weight  is nontrivial. The simplified approach adopted in our work provides an efficient way to ensure both effective adversarial interaction and training stability.
>
> |Adversary setting|Mean episode length @10k|Mean lower reward @10k|Mean upper reward @10k|
> |-|:-:|:-:|:-:|
> |weak $\pi_a^u$ + weak $\pi_a^l$|976.4|21.1|148.7|
> |weak $\pi_a^u$ + strong $\pi_a^l$|11.66|0.0|0.0|
> |strong $\pi_a^u$ + weak $\pi_a^l$|912.2|10.8|91.7|
> |strong $\pi_a^u$ + strong $\pi_a^l$|11.37|0.0|0.0|
>
> |Method|$E_\text{vel}\downarrow$|$E_\text{ang}\downarrow$|$E_\text{jpe}^\text{upper}\downarrow$|$E_\text{kpe}^\text{upper}\downarrow$|$E_\text{action}^\text{upper}\downarrow$|$E_\text{action}^\text{lower}\downarrow$|$E_\text{g}\downarrow$|$\text{Survival}\uparrow$|
> |-|:-:|:-:|:-:|:-:|:-:|:-:|:-:|:-:|
> |ALMI (Ours)|**0.2202**|**0.4812**|**0.2116**|**0.0458**| **0.0600**| **0.0175**|**0.8551**|**0.9723**|
> |weak $\pi_a^u$ + weak $\pi_a^l$|0.3415|1.4413|0.29469| 0.0526|0.0970|0.0183|1.0250|0.9612|
> |strong $\pi_a^u$ + weak $\pi_a^l$|1.1220|4.6715|2.7251| 0.1840|0.2450|4.6937|1.1702|0.3524|
> ---
> We hope these clarifications have fully addressed your concerns. If so, we would deeply appreciate your consideration in raising your score. If you have any further suggestions, please feel free to reach out.

---

> > ### Author Response · Authors · 2025-08-06
> > **Gentle Reminder**
> >
> > Dear Reviewer:
> >
> > We wanted to express our gratitude for your insightful feedback during the review process of our paper. We hope we have resolved all the concerns and showed the improved quality of our paper. Please do not hesitate to contact us if there are other clarifications we can offer.
> >
> > Best,
> >
> > The authors.

---

> ### Author Response · Authors · 2025-08-08
>
> Dear Reviewer XwBe, thanks for your thoughtful review. As the author-reviewer discussion period is near its end, we wonder if our rebuttal addresses your concerns. Please let us know if any further clarifications or discussions are needed!

---

### Official Review · Reviewer_xGSi · 2025-07-03

**Clarity:** 3
**Significance:** 3
**Originality:** 3
**Rating:** 4
**Confidence:** 4

**Summary:**

This paper presents ALMI (Adversarial Locomotion and Motion Imitation), a novel framework for humanoid control that separately trains upper-body and lower-body policies in an adversarial setup. During training, the lower body learns to perform robust locomotion under adversarial disturbances from the upper body, while the upper body learns to track motions accurately despite unstable base movement. The algorithm is validated on the full-size Unitree H1-2 robot. The authors also introduce ALMI-X, a dataset with over 80K trajectories labeled with natural language descriptions. Experiments are conducted in both simulation and real-world settings.

**Questions:**

The policies are trained and tested in Isaac Gym, yet the dataset is collected in MuJoCo. What’s the motivation or specific consideration behind using MuJoCo for ALMI-X data collection?

**Ethical Concerns:**

["NO or VERY MINOR ethics concerns only"]

**Final Justification:**

The authors have addressed my concerns in the rebuttal. I think the idea is interesting and the experimental results are reasonable.

**Limitations:**

Yes.

**Paper Formatting Concerns:**

No.

**Quality:**

2

**Strengths And Weaknesses:**

Strengths:

The core idea of treating upper-body and lower-body control as adversaries is novel and interesting. To the best of my knowledge, this is the first time such adversarial interplay has been used to encourage robustness in humanoid control.

The separation of motion roles (locomotion vs. imitation) helps stabilize policy learning.

The authors release a dataset (ALMI-X) with language annotations, which could potentially support future work in language-conditioned whole-body control.

Weaknesses:

While the adversarial framework is promising, it is inherently limited to motion decomposition (i.e., clear separation between upper- and lower-body tasks). It may not generalize well to agile whole body coordination.

The foundation model component feels disconnected from the core contribution. If I understand correctly, the model is trained in a supervised fashion on the collected ALMI-X dataset using language annotations, but the results are still far from a true "foundation model" for whole-body control.

In comparison, stronger motion generation performance might be achieved by combining diffusion-based human motion synthesis with a physics-based motion tracking controller.

The real-world robot experiments appear overly conservative, especially in locomotion behavior. The demo videos on the project page mostly showcase the ALMI policy, not the foundation model. Is that correct?

---

> ### Author Rebuttal · Authors · 2025-07-31
>
> We sincerely appreciate your thorough review and constructive feedback.
>
> **W1: Limitation of motion decomposition**
>
> We thank the reviewer for the insightful comment. We agree that a clear separation between upper and lower-body tasks may constrain the expressiveness required for agile, whole-body coordinated motions. Indeed, end-to-end policy learning over the full body’s degrees of freedom has the potential to produce more fluid and dynamic behaviors.
>  However, our work focuses on mobile manipulation with full-size humanoid robots, such as the H1-2, which involves challenging scenarios like lifting and transporting heavy objects. In such tasks, **achieving robustness is particularly critical**. We found that end-to-end policies often struggle to generalize under unseen disturbances from arm motions, which can severely affect balance. By decoupling the control of the upper and lower body, ALMI improves robustness by allowing the lower-body controller to maintain balance despite diverse, unseen upper-body behaviors.
> Additionally, the **modularity** of our framework brings practical benefits. For example, it allows the upper-body policy to be easily swapped out for a pre-trained manipulation model like VLA, enabling better generalization and task flexibility. As demonstrated in our supplementary video (e.g., door-opening tasks), ALMI is still capable of achieving coordinated behaviors across the full body, suggesting that despite the decomposition, our approach maintains a reasonable degree of whole-body integration.
> We will **revise the main text to clarify this design trade-off** and better position ALMI as a practical, robust solution for mobile manipulation, while **acknowledging its limitations for highly agile full-body coordination**.
>
> We also added two popular whole-body motion tracking methods to the baseline experiments: OmniH2O [1], Exbody2 [2] and compared them with our method on Unitree G1 robot.
>
> - **Experiment Design and Evaluation Metrics.** We adopt the CMU MoCap dataset with $1122$ motion clips to evaluate different metrics, which are same as our baseline experiment in paper, in IsaacGym. As OmniH2O and Exbody2 are imitation-based whole-body control methods without velocity tracking, so, for ALMI, we used linear and angular velocity provided by motion dataset as velocity tracking command; for the other methods, they directly track motions.
>
>  - The comparative results are shown as follows. The experimental results show that: 1) Compared with the current popular imitation-based whole-body control methods, our iterative adversarial upper and lower body method is still promising in tracking accuracy and robustness. 2) Our method has excellent scalability for the robot platform.
>
> |Method|$E_\text{vel}\downarrow$| $E_\text{ang}\downarrow$| $E_\text{jpe}^\text{upper}\downarrow$| $E_\text{kpe}^\text{upper}\downarrow$| $E_\text{action}^\text{upper}\downarrow$| $E_\text{action}^\text{lower}\downarrow$| $E_\text{g}\downarrow$|$\text{Survival}\uparrow$|
> |-|-|-|-|-|-|-|-|-|
> |ALMI (Ours)|**0.1396**|**0.2776**|**0.2367**|**0.0411**| **0.0198**|**0.7411**|**0.0977**|**0.9484**|
> |OmniH2O|0.1615|0.4166|1.0826|0.0598|1.2773|2.3219| 0.1696|0.3882|
> |Exbody2|0.4015|0.6066|0.3821|0.0719|1.1797|1.3547| 0.3367|0.8848|
>
> [1] He, Tairan, et al. "Omnih2o: Universal and dexterous human-to-humanoid whole-body teleoperation and learning." arXiv preprint arXiv:2406.08858.
> [2] Ji, Mazeyu, et al. "Exbody2: Advanced expressive humanoid whole-body control." arXiv preprint arXiv:2412.13196.
>
> **W2: Limited capacity of the foundation model**
>
> Thank you very much for pointing out this issue. As mentioned in our paper, the capabilities of the foundation model still have considerable room for improvement, and we will revise the manuscript to **tone down the claims** regarding this part accordingly.
> It is also worth noting that training a whole-body humanoid control model for language-to-action purely through supervised learning is highly challenging, as humanoid robots exhibit high degrees of freedom and are subject to frequent state discontinuities caused by intermittent foot contacts. Our experiments represent a preliminary attempt to explore this difficult yet promising direction using the ALMI-X dataset. We hope that ALMI-X and our experimental results can provide useful insights for future research in this area. However, it is important to clarify that solving this problem is not the primary focus of this study.
>
> **W3: Comparison with motion generation + motion tracking**
>
> We sincerely thank the reviewer for the thoughtful comments. The paradigm of combining diffusion-based human motion synthesis with a physics-based motion tracking controller is indeed capable of producing more diverse and expressive motions. However, current motion generation methods often do not sufficiently account for the physical constraints of the real world. In practice, these motions typically require retargeting process to adapt them to the robot’s embodiment, and directly tracking them with a motion tracking controller rarely succeeds without extensive motion repair and filtering processes which significantly hinder large-scale motion generation and real-world policy deployment. Although some existing works have achieved similar functionalities, such as Bumblebee [1], they often lack robustness—particularly in tasks requiring interaction with environment like mobile manipulation.
> In contrast, our method enables robust execution of relatively stable motions and supports large-scale generation, offering a more practical solution under real-world physical constraints.
>
> [1] Wang, Yuxuan, et al. "From Experts to a Generalist: Toward General Whole-Body Control for Humanoid Robots." arXiv preprint arXiv:2506.12779 (2025).
>
> **W4: Real-world experiments appear to be conservative**
>
> Thank you very much for the constructive feedback. (i) The mobile manipulation tasks demonstrated in our video—such as vacuuming, carrying boxes, and operating a refrigerator—involve interaction with external objects, which introduces disturbances during execution. These tasks require the robot to constantly maintain balance while performing the manipulation. Although such tasks may appear more conservative compared to highly dynamic behaviors like dancing—which are implemented using motion tracking techniques. These methods cannot interact with external objects and often suffer from poor robustness.(ii) The demonstrations shown in our video are performed using the ALMI policy rather than the foundation model. The foundation model we trained is intended as an initial exploration into the challenging and under-explored problem of language-to-action humanoid robot control, while also serving to validate the practicality of the ALMI-X dataset. As such, this part of the work remains preliminary. The model’s performance in simulation is still unstable and not yet suitable for large-scale deployment.
>
> **Q1: Motivation of collecting data in MuJoCo**
>
> Thanks for the question. According to Sec. 6 and Sec. B.5 in our paper, we follow the common practice in this field and adopt sim-to-sim transfer (i.e., Isaac to MuJoCo) to cross-validate our policy. Specifically, IsaacGym can utilize NVIDIA’s GPU resources to enable highly parallel training, while MuJoCo is specifically designed for physics simulation and offers stable and reliable results for sim-to-real transfer. As a result, we adopt MuJoCo to generate more accurate robot trajectories to facilitate sim-to-real transfer in future work.
>
> ---
> We hope these clarifications have fully addressed your concerns. If so, we would deeply appreciate your consideration in raising your score. If you have any further suggestions, please feel free to reach out.

---

> > ### Comment · Reviewer_xGSi · 2025-08-07
> >
> > Thank you for the rebuttal. My concerns are addressed, and I will change my rating to 4: borderline accept.

---

> > > ### Author Response · Authors · 2025-08-08
> > >
> > > We are pleased that our response has addressed the reviewer’s concerns. We sincerely appreciate the reviewer’s positive assessment of our work and the time dedicated to reviewing our paper.

---

> ### Author Response · Authors · 2025-08-06
> **Gentle Reminder**
>
> Dear Reviewer:
>
> We wanted to express our gratitude for your insightful feedback during the review process of our paper. We hope we have resolved all the concerns and showed the improved quality of our paper. Please do not hesitate to contact us if there are other clarifications we can offer.
>
> Best,
>
> The authors.

---

### Official Review · Reviewer_F1ym · 2025-07-04

**Clarity:** 3
**Significance:** 3
**Originality:** 3
**Rating:** 3
**Confidence:** 3

**Summary:**

This paper introduces ALMI (Adversarial Locomotion and Motion Imitation), a novel framework for humanoid robot control that addresses the challenge of whole-body coordination by separately learning policies for upper and lower body control through adversarial training. The key insight is that conventional approaches that mimic whole-body motions often fail because they don't account for the distinct roles of upper body (motion tracking) and lower body (stable locomotion). The authors formulate this as a two-player zero-sum Markov game where the lower body learns robust locomotion while the upper body tracks various motions, with each acting as an adversary to the other. They demonstrate their approach on the Unitree H1-2 robot, create a large-scale dataset (ALMI-X) with 80K+ trajectories and language annotations, and train a foundation model for end-to-end control.

**Questions:**

* How does your approach compare to recent work on language-guided robot control, particularly by Mower, Bou Ammar and others (integrated into "ROS-LLM: A ROS framework for embodied AI with task feedback and structured reasoning")? While your method uses language annotations in the dataset, ROS-LLM integrates LLMs directly into the control loop for task reasoning and feedback. Could you discuss the trade-offs between your adversarial RL approach versus LLM-based reasoning for humanoid control?

* The simplified framework samples adversarial commands rather than updating both policies. Have you experimented with true simultaneous policy updates? What are the computational and stability trade-offs?

* Why does the foundation model performance degrade on the full dataset? Have you considered alternative architectures or training strategies to address this limitation?

* How dependent is ALMI on the specific kinematics of the Unitree H1-2? Would it work with a TALOS or a G1 or a Booster? What modifications would be needed to apply it to other humanoid platforms?

* The experiments focus on relatively short motion sequences. How does the method perform on longer-horizon tasks that require extended coordination between upper and lower body?

* What are the primary failure modes of the trained policies? When does the adversarial training fail to find a good equilibrium?

**Ethical Concerns:**

["NO or VERY MINOR ethics concerns only"]

**Final Justification:**

I have adapted my scores taking the rebuttals into account!

**Limitations:**

The authors partially address limitations in Section 7, but several key limitations deserve more discussion:

- The paper doesn't discuss the computational requirements for the adversarial training process or how it scales with robot complexity.
- The approach relies on motion retargeting from human data, which may limit the types of behaviors that can be learned.
- While teleoperation is demonstrated, the paper doesn't discuss safety mechanisms for autonomous operation in human environments.
- Although successful sim-to-real transfer is shown, the specific domain randomization required suggests potential brittleness.

**Paper Formatting Concerns:**

None that I noticed!

**Quality:**

3

**Strengths And Weaknesses:**

Strengths
+ The paper includes extensive experiments in simulation and real-world deployment, demonstrating robust performance across various metrics including motion tracking accuracy and locomotion stability.
+ The method achieves impressive real-world results on the Unitree H1-2, including loco-manipulation tasks with teleoperation, showing clear practical applicability.
+ ALMI-X represents a valuable contribution to the community - a large-scale humanoid control dataset with language annotations that could enable future research.
+ Dual curriculum mechanism for adversarial motion sampling is well-designed, starting with moderate disturbances and gradually increasing intensity.

Weaknesses
- The paper acknowledges simplifying the original max-min problem by sampling adversarial commands rather than updating both policies simultaneously. This may limit the effectiveness of the adversarial training.
- The foundation model results are preliminary and show degraded performance when trained on the complete ALMI-X dataset versus subsets, suggesting scalability issues.
- The paper only compares against Exbody and ablations of their own method. Comparisons with other state-of-the-art humanoid control methods would strengthen the evaluation.
- While the method works well for the specific robot and motions tested, it's unclear how well it would generalize to different humanoid platforms or more complex manipulation tasks.

---

> ### Author Rebuttal · Authors · 2025-07-31
>
> We sincerely thank the reviewer for the thoughtful comments and valuable suggestions.
>
> **W1:  Simplification of the max-min problem by sampling adversarial commands**
>
> (i) Simplifying the max-min problem does affect the effectiveness of adversarial training. However, without this simplification, four separate policies would be required in each round of policy learning. Specifically, in round 1, when updating $\pi^u_1$, an additional adversary $\pi^l_a$ is required; when updating $\pi^l_1$, an additional adversary $\pi^u_a$ is required. It would be computationally expensive to train an additional adversary in each round. (ii) As a result, we change the inner-loop optimization of the adversary from the parameter space to the command space, which allows us to sample adversarial command to approximate the effects of training adversarial policies. (iii) ***We add additional experiments to show the necessity of the simplification***:
> **1) Train an adversary.**
> If we do not adopt the simplified approach, we would instead directly train an adversary to attack the policy. **However, it is very challenging to properly control the strength of the adversary**. Take the training of lower body policy as example, we jointly train an upper-body adversary to minimize the lower-body reward. We vary `action_clip_scale` to control adversary strength, where `action_clip` = 100 × `action_clip_scale`. The results are shown in the table below. It can be seen that large adversary collapse training, while weak adversaries yield poor robustness. In contrast, our curriculum-based method, guided by motion difficulty, provides effective and stable adversarial training.
>
> | Upper body action clip scale  | Mean episode length @20k | Mean reward @20k|
> |:-:|:-:|:-:|
> |1|7.0|0.0|
> |0.1|612.3|4.21|
> |0.01|978.5|25.79|
>
> | Method |$E_\text{vel}\downarrow$| $E_\text{ang}\downarrow$| $E_\text{jpe}^\text{upper}\downarrow$| $E_\text{kpe}^\text{upper}\downarrow$| $E_\text{action}^\text{upper}\downarrow$| $E_\text{action}^\text{lower}\downarrow$| $E_\text{g}\downarrow$|$\text{Survival}\uparrow$|
> |-|:-:|:-:|:-:|:-:|:-:|:-:|:-:|:-:|
> |ALMI (Ours)|0.2202|0.4812|0.2116|0.0458|0.0600|0.0175| 0.8551|0.9723|
> |`action_clip_scale`=0.1|1.2610|5.3445|\\|\\|\\|6.4459|1.7446|0|
> |`action_clip_scale`=0.01|1.3707|6.8632|\\|\\|\\|6.1924|2.3056|0|
>
> **2) Policy with adversarial goals.**
> Another way to implement adversarial training is to assign each policy two objectives. For this experiment, we train upper and lower policies jointly, each maximizing its own reward while minimizing the other’s. We vary adversarial strength via reward weights (e.g., **-1 for strong adversary and -0.1 for weak adversary**). Results show adversarial weight greatly affects training, and tuning it is nontrivial. Our simplified method ensures stable and effective adversarial interaction. Among converged settings, our method outperforms others in all metrics.
>
> |Adversary setting|Mean episode length @10k|Mean lower reward @10k|Mean upper reward @10k|
> |-|:-:|:-:|:-:|
> |weak $\pi_a^u$ + weak $\pi_a^l$|976.4|21.1|148.7|
> |weak $\pi_a^u$ + strong $\pi_a^l$|11.66|0.0|0.0|
> |strong $\pi_a^u$ + weak $\pi_a^l$|912.2|10.8|91.7|
> |strong $\pi_a^u$ + strong $\pi_a^l$|11.37|0.0|0.0|
>
> |Method| $E_\text{vel}\downarrow$| $E_\text{ang}\downarrow$| $E_\text{jpe}^\text{upper}\downarrow$| $E_\text{kpe}^\text{upper}\downarrow$| $E_\text{action}^\text{upper}\downarrow$| $E_\text{action}^\text{lower}\downarrow$| $E_\text{g}\downarrow$|$\text{Survival}\uparrow$|
> |-|-|-|-|-|-|-|-|-|
> |ALMI (Ours)|**0.2202**|**0.4812**|**0.2116**|**0.0458**| **0.0600**| **0.0175**|**0.8551**|**0.9723**|
> |weak $\pi_a^u$ + weak $\pi_a^l$|0.3415|1.4413|0.29469| 0.0526|0.0970|0.0183|1.0250|0.9612|
> |strong $\pi_a^u$ + weak $\pi_a^l$|1.1220|4.6715|2.7251| 0.1840|0.2450|4.6937|1.1702|0.3524|
>
> **W2: Scalability of the foundation model**
>
> Our foundation model study is an early step toward exploring direct language-to-action mapping using the ALMI-X dataset. To ensure practical deployability under high-frequency inference, we adopted a lightweight, conservative model design without architectural innovations, which may limit its capacity to fully utilize large-scale data. The observed performance drop on the full dataset reflects this trade-off. These experiments primarily serve to validate the dataset’s potential; future work will explore stronger model architectures, better data processing, and optimized training pipelines. **We will revise the paper to better convey the exploratory nature of this component and moderate our claims**.
>
> **W3: Limited baseline comparison**
>
> We have added new baseline methods: OmniH2O, Exbody2 and compared them with our method on Unitree G1 robot, due to space limits, please refer to our **response to Reviewer RhvN, W3**.
>
> **W4: Generalize to different humanoid platforms or more complex manipulation tasks**
>
> **(1)** We have included experiments on Unitree G1 robot. Please refer to the project page. Generalizing to a different robot mainly requires **(a) retargeting MoCap data to the new robot**, and **(b) making minor adjustments to the reward function** to encourage more natural gait patterns.
>
> **(2)** As for more complex manipulation tasks, our experiments’ tasks—vacuuming, door opening, and box carrying—are challenging mobile manipulations where upper-body motion affects lower-body stability. Our modular framework also allows easy swapping of the upper-body policy with pre-trained models like VLA for better generalization and flexibility.
>
> **Q1: Comparison with recent work on language-guided robot control**
>
> Thank you for pointing out this relevant line of work. ROS-LLM primarily focuses on robotic arms and does not involve whole-body humanoid control, where coordinated lower- and upper-body behaviors are essential. Our approach can complement such frameworks by providing low-level motor skills for complex whole-body tasks. Moreover, the language descriptions in ALMI-X typically correspond to holistic motion patterns, which are not easily decomposed into modular subgoals that LLMs can reason over. This limits the applicability of structured language reasoning in these contexts. Finally, as noted in prior works [1,2], current LLMs exhibit limited capacity for precise spatial reasoning—especially in generating coherent 3D motion sequences involving poses and rotations.
> [1] Spatialllm: A compound 3d-informed design towards spatially-intelligent large multimodal models. CVPR 2025
> [2] Evaluating spatial understanding of large language models. TMLR 2025
>
> **Q2:   Simultaneous policy updates**
>
> We have added experiments about the  simultaneous updating issue, please refer to the **response to W1**.
>
> **Q3: Foundation model performance on the full dataset**
>
> Due to the space limit, please refer to our **response to W2**
>
> **Q4: Generalize to different humanoid platforms**
>
> Please refer to our **response to W4**.
>
> **Q5: Performance on longer-horizon tasks**
>
> Thank you for the question. (i) For lower-body locomotion, the adversarial difficulty depends on survival length and the motion scale. Thus, longer sequences do not directly increase the coordination difficulty. (ii) For upper-body tracking, we adopt a frame-by-frame motion tracking policy; thus, the reward function can also accurately capture the tracking accuracy even with long motion sequences. In addition, the motion tracking is only conducted in the upper body, which reduces the difficulty of motion tracking. In our experiments, we find that tracking long motions does not affect the tracking performance. We’ve also tested longer AMASS motions on the G1 robot.
>
> **Q6: Primary failure modes and training failure cases**
>
> (i) **Inference-time** failure modes mainly arise from common edge cases in humanoid robot control, such as carrying excessively heavy loads or unexpected actuation delays, which may exceed the training distribution. (ii) During **training**, adversarial learning without curriculum often leads to unstable training process. Moreover, insufficient training iterations may prevent the upper and lower body policy from reaching a stable equilibrium.
>
> **L1: Computational requirements and how it scales with robot complexity**
>
> We report the training time of ALMI for both H1-2 and G1 robots, all trained on an NVIDIA RTX 4090, as shown in table below.
>
> |ALMI (H1-2)|ALMI (G1)|
> |-|-|
> |Lower-1\~3: 10k, 6k, 5k iterations with 2.5s/iter; Upper-1\~2: 3k, 1k iterations with 2.0s/iter;  **Total: about 16.8 hours**|Lower-1\~3: 10k, 8k, 6k iterations with 2.5s/iter; Upper-1\~2: 3k, 3k iterations with 2.0s/iter;  **Total: about 20.0 hours**|
>
> **L2: Motion retargeting from human data**
>
> Thank you for the comment. Although our method relies on motion retargeting from human data, the use of the diverse AMASS dataset ensures good generalization. We also validated the policy on the real H1-2 robot via teleoperation, showing it can mimic various upper-body motions, suggesting that the human data dependency does not significantly limit behavior diversity.
>
> **L3: Safety mechanisms**
>
> Thank you for the suggestion. Given H1-2's whole-body scale, safety is crucial. We enforced safe joint velocity and workspace limits for all end-effectors and included an emergency stop to disable the policy when needed.
>
> **L4: About domain randomization**
>
> We acknowledge the concern. Domain randomization helps bridge the sim-to-real gap by compensating for imperfect simulation and real-world variations like torque noise and environment changes, making it essential for robust transfer [1].
> [1] Peng, Xue Bin, et al. Sim-to-real transfer of robotic control with dynamics randomization. ICRA, 2018
>
> ---
> We hope these clarifications have fully addressed your concerns. If so, we would deeply appreciate your consideration in raising your score. If you have any further suggestions, please feel free to reach out.

---

> ### Author Response · Authors · 2025-08-08
>
> Dear Reviewer F1ym, thanks for your thoughtful review. As the author-reviewer discussion period is near its end, we wonder if our rebuttal addresses your concerns. Please let us know if any further clarifications or discussions are needed!

---

### Official Review · Reviewer_RhvN · 2025-07-13

**Clarity:** 3
**Significance:** 3
**Originality:** 3
**Rating:** 5
**Confidence:** 3

**Summary:**

This paper presents the ALMI framework, which introduces an adversarial learning paradigm for training upper and lower-body policies in humanoid control. The key idea is to formulate the coordination between upper and lower body as a two-player zero-sum game, thereby encouraging robustness and precision in full-body motion generation. The method is validated through comprehensive experiments both in simulation and on the real H1 humanoid robot. Additionally, the authors contribute a large-scale language-annotated dataset for whole-body control, which holds promise for advancing foundation policy learning in robotics.

**Questions:**

1. In Lines 195–196, the adversarial signal to the upper-body policy consists solely of velocity commands from the lower body. Could the authors clarify why this information alone is sufficient? For instance, would incorporating additional signals such as gait type, gait frequency, or phase provide stronger adversarial influence and improve learning outcomes?

**Ethical Concerns:**

["NO or VERY MINOR ethics concerns only"]

**Final Justification:**

I maintained my score of 5, appreciating the contribution of the work.

**Limitations:**

yes

**Paper Formatting Concerns:**

No concern.

**Quality:**

3

**Strengths And Weaknesses:**

## Strengths
1. Novel adversarial learning design: The proposed adversarial formulation for upper-lower body coordination is a promising and innovative direction. Framing the interaction as a two-player zero-sum game, combined with curriculum learning for the lower body, introduces a structured and scalable approach to whole-body policy learning.
2. Real-world deployment: The evaluation on the H1 humanoid robot in real-world settings is a major strength, demonstrating the practicality and robustness of the proposed method beyond simulation.
3. Good writing and experimental design: The experiments are well-structured, with evaluation metrics covering motion tracking accuracy, command fidelity, stability, and survival rate. The ablation studies effectively highlight the benefits of different modules within the framework.
4. Open-source contributions: The release of code and a large-scale dataset with language annotations adds substantial value to the community and paves the way for future research in language-conditioned whole-body control.

## Weaknesses
1. Reward design in Appendix: The reward structure, which plays a critical role in training, is relegated to the appendix. Including a concise version of it in the main text would greatly benefit the reader's understanding of the method.
2. Symbol clarity: The preorder symbol introduced in Line 117 should be clearly defined to avoid ambiguity.
3. Limited baseline comparison: The experiments compare only against Exbody. Including additional baseline methods, such as recent approaches in modular or hierarchical control, would strengthen the empirical validation.
4. Minor typos:
   - Line 90: The symbol should be \( a^u \) instead of \( a_l \).
   - Line 159: The citation should refer to Eq. (1) instead of Eq. (2).

---

> ### Author Rebuttal · Authors · 2025-07-31
>
> We sincerely thank the reviewer for the positive assessment of our work.
>
> **W1: Reward design in Appendix**
>
> Thank you for the suggestion. We will consider moving important components such as the reward design and curriculum settings to the main text to aid readers' understanding.
>
> **W2: Symbol clarity about the preorder symbol introduced in Line 117**
>
> Thanks for the suggestion, we will add the definition of the preorder symbol in the paper. The symbol $a\asymp b$ means two variables have the same order of magnitude. Specifically, there exist positive constants $C_1$ and $C_2$ such that for sufficiently large values of the relevant variable, we have $C_1 \cdot a \leq b \leq C_2 \cdot a$ holds.
>
> **W3: Limited baseline comparison**
>
> Thank you for the suggestion. To indicate that our method can be extended to other robot platforms, we added new baseline methods: OmniH2O [1], Exbody2 [2] and compared them with our method on Unitree G1 robot.
>
> - **Experiment Design and Evaluation Metrics.** We adopt the CMU MoCap dataset with $1122$ motion clips to evaluate different metrics, which are same as our baseline experiment in paper, in IsaacGym. As OmniH2O and Exbody2 are imitation-based whole-body control methods without velocity tracking, so, for ALMI, we used linear and angular velocity provided by motion dataset as velocity tracking command; for the other methods, they directly track motions.
>
> - **Training Details and Checkpoint Selection.** All the policies are trained on one NVIDIA RTX4090. The training time and the checkpoint of each method selected in the baseline experiment are shown in the following table:
>
> 	|             | Training Time Consumption                                    | Experiment Checkpoint             |
> 	| ----------- | ------------------------------------------------------------ | --------------------------------- |
> 	| ALMI (ours) | Lower-1～3: 10k, 8k, 6k iterations with 2.5s/iter; Upper-1～2: 3k, 3k iterations with 2.0s/iter. **Total: about 20.0 hours** | Lower-3 6k iter + Upper-2 3k iter |
> 	| OmniH2O     | Teacher: 30k iterations with 1.5s/iter; Student: 10k iterations with 1.5s/iter; **Total: about 16.6 hours** | Student 10k iter                  |
> 	| Exbody2     | Teacher: 20k iterations with 2.5s/iter; Student: 4k iterations with 2.0s/iter; **Total: about 16.1 hours** | Student 4k iter                   |
>
>
>  - The comparative results are shown as follows. The experimental results show that: 1) Our method has excellent scalability for the robot platform. 2) Compared with the current popular imitation-based whole-body control methods, our iterative adversarial upper and lower body method is still promising in tracking accuracy and robustness.
>
> |Method|$E_\text{vel}\downarrow$| $E_\text{ang}\downarrow$| $E_\text{jpe}^\text{upper}\downarrow$| $E_\text{kpe}^\text{upper}\downarrow$| $E_\text{action}^\text{upper}\downarrow$| $E_\text{action}^\text{lower}\downarrow$| $E_\text{g}\downarrow$|$\text{Survival}\uparrow$|
> |-|-|-|-|-|-|-|-|-|
> |ALMI (Ours)|**0.1396**|**0.2776**|**0.2367**|**0.0411**| **0.0198**|**0.7411**|**0.0977**|**0.9484**|
> |OmniH2O|0.1615|0.4166|1.0826|0.0598|1.2773|2.3219| 0.1696|0.3882|
> |Exbody2|0.4015|0.6066|0.3821|0.0719|1.1797|1.3547| 0.3367|0.8848|
>
> [1] He, Tairan, et al. "Omnih2o: Universal and dexterous human-to-humanoid whole-body teleoperation and learning." arXiv preprint arXiv:2406.08858.
> [2] Ji, Mazeyu, et al. "Exbody2: Advanced expressive humanoid whole-body control." arXiv preprint arXiv:2412.13196.
>
> **W4: Minor typos**
>
> Thank you for the typo corrections, we will fix them.
>
> **Q1: Additional lower body signals for stronger adversarial influence**
>
> That is an insightful question. We agree that incorporating richer gait-related signals could strengthen the adversarial influence. However, the adversarial signal depends on the input command and capability of the lower-body controller. Since our main target is mobile manipulation task, we did not consider diverse gait types for lower body. We plan to explore this direction in future work, potentially by integrating with frameworks like HugWBC [1].
>
> [1] Xue, Y.; Dong, W.; Liu, M.; Zhang, W.; and Pang, J. 2025. A Unified and General Humanoid Whole-Body Controller for Fine-Grained Locomotion. In Robotics: Science and Systems (RSS).
>
> ---
> We hope these clarifications have fully addressed your concerns. If you have any further suggestions, please feel free to reach out.

---

> > ### Author Response · Authors · 2025-08-06
> > **Expectations for Addressing Reviewer Concerns**
> >
> > Dear Reviewer,
> >
> > We wanted to express our gratitude for your insightful feedback during the review process of our paper. We hope we have satisfactorily addressed all your concerns and demonstrated the improved quality of our work. If there are any additional points you'd like us to consider, please let us know. Your insights are invaluable to us, and we're eager to address any remaining issues.
> >
> > Thank you for your time and effort in reviewing our paper.
> >
> > Best regards,
> >
> > The authors

---

> > ### Comment · Reviewer_RhvN · 2025-08-06
> >
> > Thank you to the authors for the detailed and thoughtful rebuttal. I appreciate the effort to address each point raised in the initial review.
> >
> > Most of my concerns have been adequately resolved, and I continue to view this work as a well-executed and impactful contribution to whole-body humanoid control. I therefore maintain my score of 5 (Accept).

---

> > > ### Author Response · Authors · 2025-08-08
> > >
> > > We are pleased that our response has addressed the reviewer’s concerns. We sincerely appreciate the reviewer’s positive assessment of our work and the time dedicated to reviewing our paper.

---

### Note · Authors · 2025-08-12

**Dear AC and Reviewers,**

We sincerely appreciate your dedication to the conference and the valuable time and effort you have invested in reviewing our manuscript.

Our paper introduces **Adversarial Locomotion and Motion Imitation (ALMI)**, a novel framework for adversarial policy learning between the upper and lower body of humanoids. As the reviewers kindly noted, our method demonstrates: Strong theoretical motivation (RhvN, F1ym), insightful contributions and a practical algorithm (RhvN, F1ym, XwBe), and significant empirical improvements (RhvN, xGSi).

We deeply appreciate your constructive feedback and we have carefully added detailed and comprehensive explanation and experiments in the rebuttal as follows:
1. **New Baseline Comparisons:**
    - Experiments on **Exbody2** and **OmniH2O**, representing state-of-the-art imitation-based whole-body control methods.
2. **Supplementary Experiments and Theoretical Validation:**
    - Ablation studies on **the original theory implementation** and empirical verification that our **command-based curriculum** simplifies the theory while enhancing practicality and efficiency.
    - Ablation studies on **simultaneous upper/lower-body policy updates**, further validating the superiority of our adversarial iterative approach.
    - Analysis of **automatic policy updates** during training, assessing their trade-offs.
3. **Additional Clarifications:**
    - **Scalability** across different robot platforms.
    - **Training efficiency**, demonstrating negligible overhead in time and resources.
    - **Foundation model comparisons**: Direct text-to-action vs. motion generation + tracking.

Meanwhile, we have carefully and thoroughly answered each of the reviewers' questions. We believe that the comprehensive experimental design and results and complete theoretical analysis provided in our rebuttal sufficiently address the reviewers' points. Their positive feedback has further strengthened our confidence, for which we are sincerely grateful. We kindly request that the AC dedicate more attention to the evaluation of our submission. We kindly request the AC’s further evaluation of our submission.

**Thank you very much,**
The Authors

---

### Decision · Program_Chairs · 2025-09-17

**Decision:**

Accept (poster)

**Comment:**

This paper tackles the problem of learning a whole-body controller by decomposing the problem into learning separate controllers for lower- and upper-body motion. This learning setup is done through an adversarial optimziation process as a zero-sum game: the locomotion/lower-body policy seeks to maximize the ability to follow locomotion commands, and the upper-body controller seeks to minimize disturbances. The lower-body must also account for potentially adversarial upper-body motions. The reviewers agree with the authors' claim that the approach is novel and that the dataset is interesting. The real-world deployment and positive results were also strengths. Reviewers had concerns that important technical details were relegated to the appendix. Further, the paper did not adequately benchmark against state-of-the-art baselines. The authors appear have succesfully rebutted the benchmarking concern by adding baselines and showing positive reuslts. If this is incorporated into the paper, that would be helpful. Overall, the paper seems to make a contribution to the whole-body locomotion literature even if the adversarial setup may not fully take advantage of the complete capabilities of such a system (i.e., cooperative optimization).